# Parameter Efficient Fine-Tuning of Segment Anything Model for Biomedical Imaging

**Carolin Teuber**[*1,2] [iD]                    CAROLIN.TEUBER@STUD.UNI-GOETTINGEN.DE
**Anwai Archit**[*2] [iD]                              ANWAI.ARCHIT@UNI-GOETTINGEN.DE
**Constantin Pape**[2,3,4] [iD]        CONSTANTIN.PAPE@INFORMATIK.UNI-GOETTINGEN.DE

[1] *Georg-August-University Göttingen, Institute of Physics*

[2] *Georg-August-University Göttingen, Institute of Computer Science*

[3] *CAIMed - Lower Saxony Center for AI & Causal Methods in Medicine, Göttingen*

[4] *Cluster of Excellence Multiscale Bioimaging (MBExC), Georg-August-University Göttingen*

**Editors:** Accepted for publication at MIDL 2025

## Abstract

Segmentation is an important analysis task for biomedical images, enabling the study of individual organelles, cells or organs. Deep learning has massively improved segmentation methods, but challenges remain in generalization to new conditions, requiring costly data annotation. Vision foundation models, such as Segment Anything Model (SAM), address this issue through improved generalization. However, these models still require finetuning on annotated data, although with less annotations, to achieve optimal results for new conditions. As a downside, they require more computational resources. This makes parameter-efficient finetuning (PEFT) relevant. We contribute the first comprehensive study of PEFT for SAM applied to biomedical images. We find that the placement of PEFT layers is more important for efficiency than the type of layer for vision transformers and we provide a recipe for resource-efficient finetuning.

**Keywords:** segment anything, peft, biomedical image segmentation

## 1. Introduction

Segmentation is a fundamental analysis task for biomedical images. It enables the study of individual objects, such as cells and organelles in microscopy, or organs and lesions in medical imaging. Deep learning has massively advanced the field. However, the large diversity of modalities and tasks so far required different methods for specific applications, such as CellPose (Pachitariu and Stringer, 2022) and Stardist (Schmidt et al., 2018) for cell and nucleus segmentation, or nnU-Net (Isensee et al., 2020) and TotalSegmentator (Wasserthal et al., 2023) for segmentation in radiology. Adapting any of these models to a new modality or a new task requires further data annotation for training due to limited generalization. Annotations can only be provided by experts and are time-consuming, making adaptation costly and preventing the wider use of automatic segmentation.

Vision foundation models for image segmentation, e.g. Segment Anything Model (SAM) (Kirillov et al., 2023) or SEEM (Zou et al., 2023), promise a more unified solution. These models have been trained on large annotated datasets and address both interactive and

---

[*] Contributed Equally

automatic segmentation. They were recently adapted to biomedical images, resulting in foundation models for microscopy (Archit et al., 2025a; Israel et al., 2024) and medical imaging (Ma et al., 2024; Zhao et al., 2024; Archit et al., 2025b). These models are applicable to many tasks in their respective domain without adaptation, but specific finetuning can further improve them (Archit et al., 2025a). Notably, they need fewer annotated data compared to other models, requiring as few as a single labeled image (Zhou et al., 2024).

Most vision foundation models use a vision transformer (ViT) (Dosovitskiy et al., 2021) as encoder. Hence, they typically have more parameters than previous architectures, making training more resource demanding and requiring a high-end GPU, even for small training sets. To enable efficient adaptation, parameter-efficient finetuning (PEFT) has emerged, for example through low rank adaptation of attention layers (LoRA) (Hu et al., 2022). Instead of updating all model parameters, these methods either update only a small subset of parameters, or they introduce a few new parameters that are updated, while freezing the rest of the model. While PEFT has been extensively studied for large language models (Pu et al., 2023; Xu et al., 2023; Balne et al., 2024), its application in computer vision, particularly for segmentation tasks, is less explored. Several approaches that adapt SAM to biomedical segmentation use PEFT, e.g. (Archit et al., 2025a; Zhou et al., 2024) for microscopy and (Gu et al., 2025; Zhang and Liu, 2023; Wei et al., 2024; Wu et al., 2025) for medical imaging. However, they primarily use LoRA without investigating its hyperparameters or alternatives. To our knowledge, none of this prior work studied the practical efficiency gains through PEFT, relying on the trainable parameter count as a measure for efficiency. Other prior work has studied PEFT for classification in natural images (Xin et al., 2024) as well as classification and text-to-image generation in medical images (Dutt et al., 2024).

We address this gap by studying PEFT for biomedical segmentation; for SAM and its variants $\mu$SAM (Archit et al., 2025a) and MedicoSAM (Archit et al., 2025b). We contribute:

- An evaluation of several PEFT methods on microscopy and medical imaging.

- A thorough evaluation of efficiency gains due to PEFT.

- Implementation of more efficient "late PEFT" approaches based on these finding and an implementation of quantized LoRA (QLoRA) (Dettmers et al., 2023) for ViTs.

- Recommendations for the use of PEFT and a recipe for efficient adaptation of SAM.

- A detailed ablation of hyperparameters for several PEFT methods, including LoRA.

Our workflow and improvements from efficient adaptation are shown in Fig. 1. We believe that our study will facilitate the use of foundation models for biomedical image segmentation. Further, it will inform future developments of PEFT for computer vision. Our code is available at https://github.com/computational-cell-analytics/peft-sam.

## 2. Methods

### 2.1. Additive and Selective Finetuning

We distinguish two types of methods: additive and selective PEFT. Selective methods update a subset of the model parameters, while freezing the rest. A simple example is

Figure 1: Overview of PEFT-SAM. a) We study PEFT of domain-specific SAMs to enable interactive correction and efficient finetuning for improved segmentation. b) Segmentation results for three datasets from $\mu$SAM, $\mu$SAM finetuned with LoRA and CellSeg1, all trained on two images (Zhou et al., 2024).

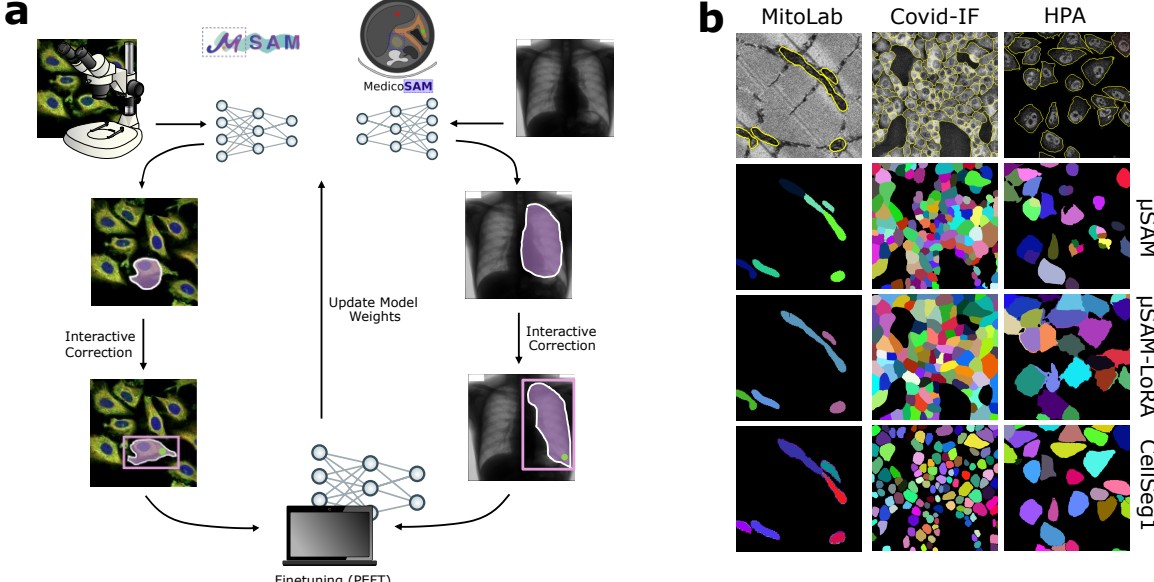

freezing parts of the model, for example SAM's image encoder, mask decoder, or prompt encoder. Freezing the image encoder, which contains most parameters, yields the greatest efficiency gain. (Archit et al., 2025a) have shown that this approach does not have a large negative impact on segmentation quality after finetuning. We refer to this approach as Freeze Encoder. The other selective PEFT methods we study are LayerNorm Tuning (LN Tune) (Basu et al., 2024), Bias Tuning (Bias Tune) (Cai et al., 2020), and Attention Tuning (Attn Tune). (Touvron et al., 2022), which update only the parameters of the normalization layers, the biases, or the attention weights, respectively.

Additive PEFT methods introduce a few additional parameters, while freezing the existing ones. Among these methods, LoRA (Hu et al., 2022) reduces the number of parameters via a low rank decomposition of the attention weight matrix:

$$\mathbf{W} = \mathbf{W}_{pretrained} + \alpha \mathbf{AB}, \tag{1}$$

where $\mathbf{A} \in \mathbb{R}^{d \times r}$, $\mathbf{B} \in \mathbb{R}^{r \times d}$. Here, the rank $r$ is much smaller than the original dimension $d$ of $\mathbf{W}$. $\alpha$ scales the learned weights. Further efficiency gains can be obtained by quantizing the frozen weights, i.e. $W_{pretrained}$ and others, to 4 bit during training, as in QLoRA (Dettmers et al., 2023). We adopt this approach for ViTs, providing, to our knowledge, its first application to this architecture; see also App. B. AdaptFormer (Chen et al., 2022) adds a lightweight module, consisting of two fully connected layers, in parallel to the MLP layers of the ViT. The outputs of the adapter are scaled by a factor $\alpha$ and added to the original MLP outputs. Scale-Shift Features (SSF) (Lian et al., 2022) performs a linear

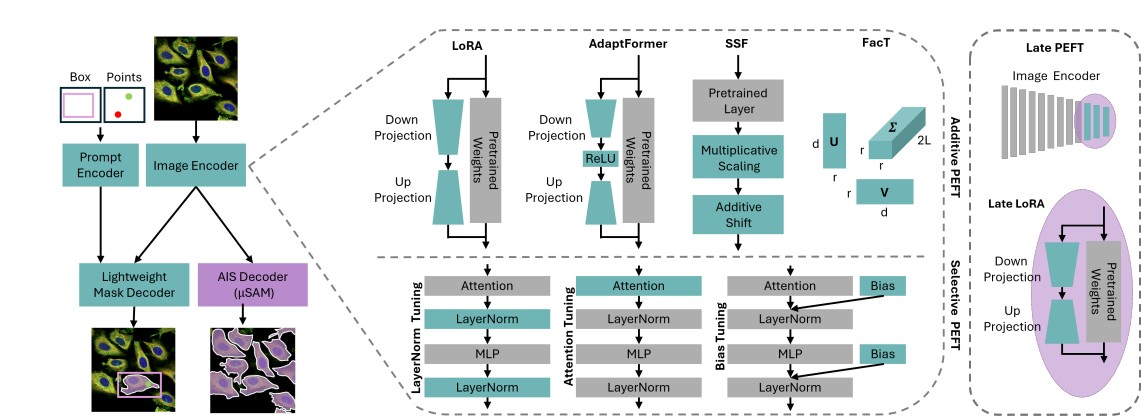

Figure 2: The original architecture of SAM, comprising image encoder, mask decoder and prompt encoder, with an additional segmentation decoder, proposed by $\mu$SAM. The image encoder is fine-tuned through additive PEFT, which introduces a few additional parameters, or selective PEFT, which updates only selected parameters. We also introduce late PEFT, which adapts only layers at the end of ViT.

transformation, introducing a scale and a shift parameter to the output of each layer in the transformer block. Factor Tuning (FacT) (Jie and Deng, 2023) combines multiple weight matrices into a single tensor and then applies a low rank decomposition to it. Here, we adopt the implementation of (Chen et al., 2024a), which tensorizes the attention weight increment matrices (corresponding to $A, B$ in Eq. 1) and then decomposes the tensor according to:

$$\Delta \mathbf{W} = \mathbf{U}\mathbf{\Sigma}\mathbf{V}^T \,, \tag{2}$$

where $\mathbf{U} \in \mathbb{R}^{d \times r}$, $\mathbf{V} \in \mathbb{R}^{d \times r}$ and $\Sigma \in \mathbb{R}^{r \times r}$. The factors $\mathbf{U}$ and $\mathbf{V}$ are shared across layers.

Fig. 2 shows an overview of these 9 PEFT methods. We apply them only to the image encoder; prompt encoder and mask decoder are always updated. Further, we study an approach we call "late PEFT", where we insert trainable parameters / unfreeze weights only in the late layers of the image encoder. This is motivated by our findings on computational efficiency, see App. A. Specifically, we study "late LoRA", "late QLoRA" and "late finetuning". We compare all PEFT methods with normal parameter updates (Full FT). We use SAM (Kirillov et al., 2023) and two domain-specific models, $\mu$SAM (Archit et al., 2025a) for microscopy, and MedicoSAM (Archit et al., 2025b) for medical imaging, to initialize weights. The latter two models have finetuned SAM on a large annotated dataset for the respective domain.

## 2.2. Interactive Segmentation

SAM has introduced a novel formulation for interactive segmentation, where users can provide input prompts, points (positive or negative), a bounding box or a rough mask, to identify an object. The model then predicts the corresponding mask by processing

the image with its image encoder, a ViT, the prompts with its prompt encoder, and the outputs of image encoder and prompt encoder with its mask decoder. Model predictions can be corrected by providing further prompts.

SAM is trained on a large dataset with annotations using an objective that simulates interactive segmentation. In each iteration, this objective first samples prompts from an annotated mask, predicts the object mask, and then iteratively corrects the prediction with additional prompts sampled from the annotation. Predictions and annotations are compared with a loss function for each iteration, and the average loss is used to update parameters. We use the implementation of this procedure from (Archit et al., 2025a).

To evaluate interactive segmentation, we automatically derive prompts from annotations to segment the object and then iteratively correct it, similar to the training objective. We perform 7 correction iterations. We compute the metric (App. D) between annotations and predictions for the initial prompt and corrections. We report the results for an initial point prompt, an initial box prompt, and the last correction iteration, when starting from a point ($I_P$, appendix only), and when starting from a box ($I_B$, appendix only).

### 2.3. Automatic Instance Segmentation

For automatic instance segmentation (AIS) we use the implementation of $\mu$SAM, which adds a UNETR decoder (Hatamizadeh et al., 2022) to SAM to predict outputs for instance segmentation. The decoder consists of four blocks, with two convolutional layers and a transposed convolution for upsampling each. The blocks also receive the image encoder's output as input. The decoder predicts three channels: the distance to the object center, the distance to the object boundary, and foreground probabilities. During training, it is updated jointly with the rest of SAM, using the annotations to derive targets for its outputs.

During inference, the segmentation is computed by deriving seeds from the distance predictions, a mask from foreground predictions, and applying a seeded watershed inside this mask. For evaluation, the instance segmentation is compared with annotations using the mean segmentation accuracy, see App. D.

### 2.4. Data

We use 6 microscopy and 6 medical imaging datasets. For light microscopy (LM), we use Covid-IF (Pape et al., 2021) with 49 immunofluorescence microscopy images and cell annotations, HPA (Ouyang et al., 2019) with 276 confocal microscopy images and cell annotations; using the channel that stains cytosol. We use GoNuclear (Vijayan et al., 2024) with 5 fluorescence microscopy and nucleus annotations. We use OrgaSegment (Lefferts et al., 2024) with 231 brightfield microscopy images and organoid annotations. We use two electron microscopy (EM) datasets: Platynereis (Vergara et al., 2021) with 3 EM volumes and cilia annotations, and a subset of MitoLab (Conrad and Narayan, 2023) with a volume of glycolytic muscles and mitochondria annotations. For the 3D datasets (GoNuclear, Platynereis, MitoLab), we perform 2D segmentation by treating slices as individual images.

For medical imaging, we use AMD-SD (Hu et al., 2024) with 3049 optical coherence tomograms and annotations for lesions. We use JSRT (Shiraishi et al., 2000) with 247 images and lung and heart annotations. We use Mice TumSeg (Jensen et al., 2024) with 452 micro-CT volumes and tumor annotations. We use Papila (Kovalyk et al., 2022) with

488 fundus images and optic cup annotations. We use MOTUM (Gong et al., 2024) with 64 brain MRI volumes and tumor annotations. We use PSFHS (Chen et al., 2024b) with 1358 ultrasound images and fetal head and pubic symphysis annotations. For the 3D datasets (MOTUM and Mice TumSeg), we perform 2D segmentation (see above).

## 3. Results

### 3.1. Comparison of PEFT methods

We evaluate the PEFT methods (Sec. 2.1) on 6 microscopy and 6 medical datasets (Sec. 2.4), using the complete datasets with a train / test split. We use SAM as base model in all cases, $\mu$SAM (either LM or EM model) for microscopy, and MedicoSAM for medical data. For microscopy we evaluate interactive and automatic segmentation via AIS (Sec. 2.3), for medical data we only evaluate interactive segmentation. Models are trained with a batch size of 2 and 25 objects per image, using a A100 GPU with 80GB of VRAM.

Figure 3: PEFT results for microscopy (left) and medical (right) segmentation. Methods are ordered by parameter count, SAM and $\mu$SAM / MedicoSAM are used as base models. Circles show the best three results per dataset / task. See also Fig. 8 and Fig. 9.

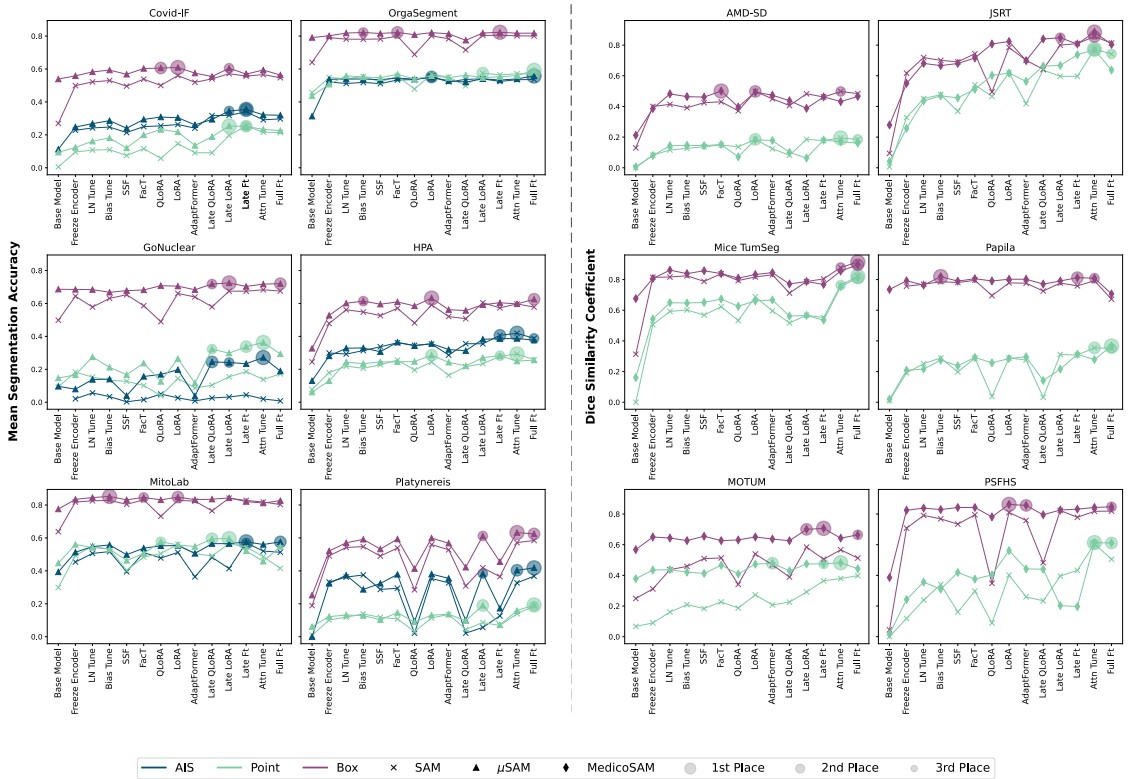

The results are shown in Fig. 3. Finetuning always improves, with the magnitude depending on the task / dataset and the base model. Full FT yields the most consistent improvement, with Attention Tuning, LoRA, Late LoRA and Late FT typically improving similarly. The improvements from other PEFT methods are not quite as consistent. Interestingly, (Late) QLoRA shows worse performance with SAM as base model, but not for $\mu$SAM / MedicoSAM. The results are summarized and discussed in more detail in App. E.1, including results for further segmentation tasks ($I_B$, $I_P$) and further medical datasets. We perform ablation studies for LoRA, Adaptformer and FacT in App. G.

We also evaluate the computational efficiency of PEFT methods. The detailed results on efficiency are given in App. E.2. Our most notable finding is that our tasks are constrained by the size of activations rather than trainable parameter count, see App. A. Consequently, most prior PEFT methods yield only a marginal efficiency gain, with the exception of freezing the encoder. We introduced "late LoRA", "late QLoRA", and "late finetuning" (late FT) due to this observation. They achieve meaningful efficiency gains while improving segmentation quality compared to full freezing of the encoder, see the respective results in Fig. 3. We analyze different late PEFT settings in detail in Fig. 4 and chose the best parameter settings based on these results for other experiments.

## 3.2. Resource-efficient Finetuning

Building on the evaluation of PEFT methods, we propose resource-efficient finetuning by adopting a similar workflow to CellSeg1 (Zhou et al., 2024), who finetune SAM for automatic segmentation on a single image using LoRA. We compare their method with our finetuning approach, explained in detail in App. C for automatic *and* interactive segmentation, using Freeze Encoder, QLoRA, LoRA (including late PEFT settings for all 3), and Full Finetuning, which offer the best efficiency-quality trade-offs. To reduce hardware demand, we lower the number of objects per image to 5, otherwise using the same settings as in Sec. 3.1. We use one image for training, one for validation, and the same test sets as before. Fig. 4 shows the results for microscopy and medical data (only interactive segmentation for the latter). SAM improves for all tasks, $\mu$SAM mainly for AIS. Our approach is on par with CellSeg1, for which we also use both base models, while also improving interactive segmentation. Interestingly, full finetuning is on par with PEFT, despite the small training data and having more trainable parameters. For large domain shifts (Platynereis) it outperforms PEFT. We also benchmark efficiency, see Tabs. 8, 9, 10. Here, we again see major gains when freezing the encoder, small gains from other PEFT, and a good trade-off using "late PEFT". Further, we investigate the effect of adding more training images in Fig. 12.

## 4. Discussion

We conducted a thorough study of PEFT for segmentation in biomedical images with SAM. Our experiments show that PEFT achieves a similar quality as full finetuning. Contrary to observations made by others, e.g. (Dutt et al., 2024), it does not lead to better results for limited training data compared to full FT. The performance of PEFT methods varies across datasets, LoRA is the best overall, QLoRA is good for small domain gaps. Efficiency gains are marginal for standard PEFT, with the exception of encoder freezing, due to the fact that our tasks are activation rather than parameter bound. Based on these findings

Figure 4: Resource-efficient training for microscopy and medical segmentation. We compare several PEFT methods in data limited settings. Our methods use one image for training, one for validation, CellSeg1 uses only a single image.

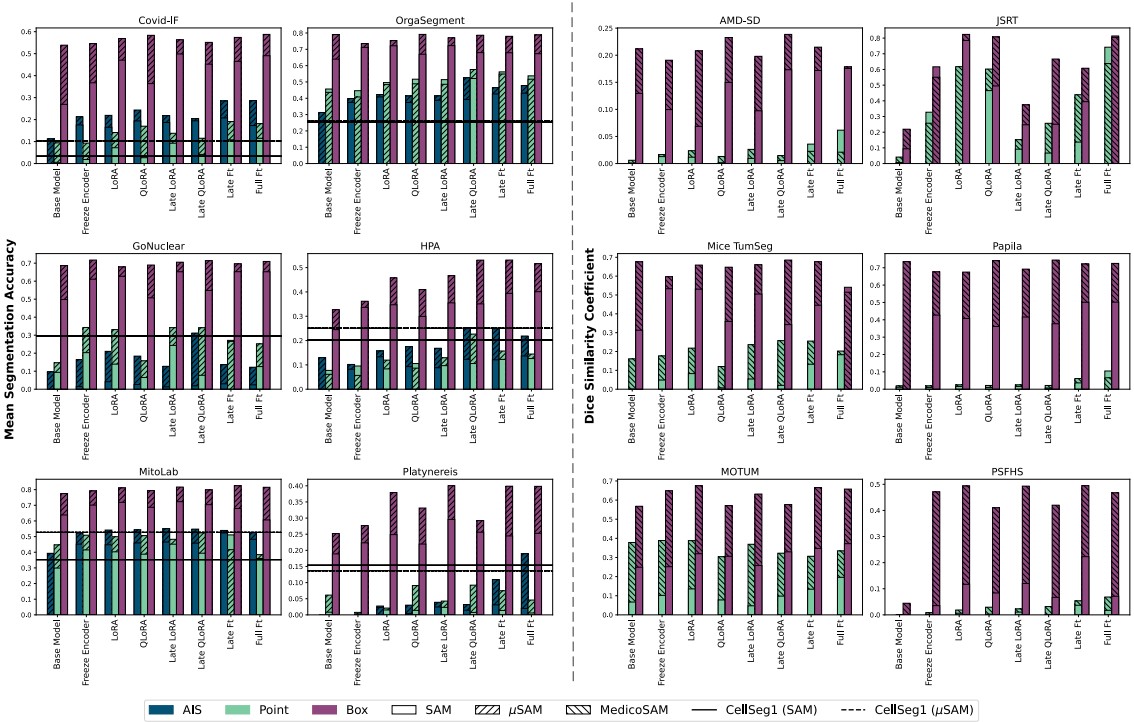

we introduce late PEFT, which improves efficiency with only small loss of quality. Given these results, we recommend freezing the encoder in very limited resource settings (e.g. on a CPU), using late PEFT for limited resources (e.g. consumer GPU), and otherwise using full finetuning (high-end GPU). See App. E.3 for more detailed recommendations.

We proposed an approach for resource-efficient finetuning, which can improve SAM with only two annotated image, similar to (Zhou et al., 2024), but also improves interactive segmentation. We believe that this approach will speed up many practical segmentation tasks, by enabling finetuning SAM previously hindered by resource demands. We have integrated this approach in the $\mu$SAM tool, which provides interactive data annotation, enabling efficient human-in-the-loop annotation and training in one tool.

We believe that our work will benefit future work on PEFT for (vision) foundation models, e.g. for SAM2 (Ravi et al., 2025). Our results on late PEFT show how such strategies need to be adapted for vision transformers, which are activation rather than parameter bound, offering a clear direction for future research.

Figure 5: a) Parameter count vs. memory requirements during training for Late LoRA, Late QLoRA and late finetuning (Late FT) for microscopy finetuning on HPA. We report two different LoRA approaches, where adapters are applied only to the attention layer (LoRA-C) or also to the MLP (LoRA-A). b) Same as a., but for finetuning on medical data (PSFHS). Note that the overall memory requirement is significantly lower because we do not train the encoder for automatic segmentation, which requires an additional forward and backward pass during training. c) Segmentation quality as a function of late finetuning on HPA and PSFHS.

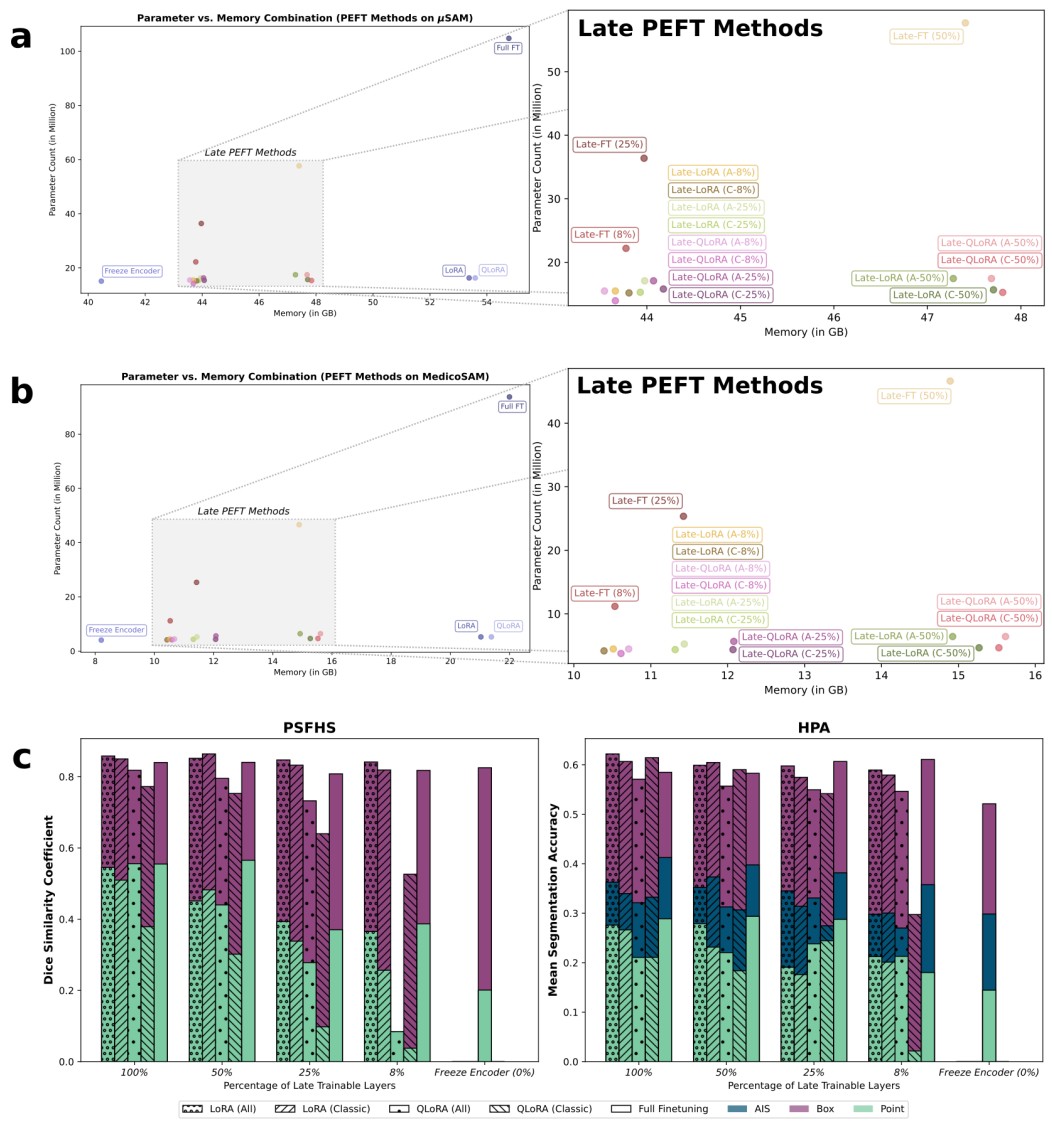

## Acknowledgments

The work of Anwai Archit was funded by the Deutsche Forschungsgemeinschaft (DFG, German Research Foundation) - PA 4341/2-1. Constantin Pape is supported by the German Research Foundation (Deutsche Forschungsgemeinschaft, DFG) under under Germany's Excellence Strategy - EXC 2067/1-390729940. This work is supported by the Ministry of Science and Culture of Lower Saxony through funds from the program zukunft.niedersachsen of the Volkswagen Foundation for the 'CAIMed – Lower Saxony Center for Artificial Intelligence and Causal Methods in Medicine' project (grant no. ZN4257). It was also supported by the Google Research Scholarship "Vision Foundation Models for Bioimage Segmentation". We gratefully acknowledge the computing time granted by the Resource Allocation Board and provided on the supercomputer Emmy at NHR@Göttingen as part of the NHR infrastructure, under the project nim00007. We acknowledge the use of icons from Bioicons (Jug and Contributors, 2023) in Fig. 1. In particular, the microscope icon by DBCLS (for Life Science , DBCLS) is licensed under CC-BY 4.0.

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

## Appendix A. Efficiency of PEFT

For the optimal use of PEFT methods in vision transformers, it is important to understand the main contributors to the memory footprint during training. Unlike typical LLM training, where trainable parameters dominate memory usage, training SAM (and likely other ViT applications) is primarily constrained by activation gradients. The distinction becomes evident if we consider the ratio of parameters and sequence length, which controls the size of activations. To illustrate this, consider the following: GPT-3, with a sequence length of 2048 tokens and 175B parameters (Brown et al., 2020), yields a ratio of approximately 85.5M. LLaMa-2 (Touvron et al., 2023), with a sequence length of 4096 and 70B parameters has a ratio of 17M. BERT-base, used with a sequence length of 512 and 110M parameters (Devlin et al., 2019), results in a ratio of 215,000. In contrast, for the Segment Anything Model (SAM), an input image size of 1024x1024 (Kirillov et al., 2023), divided into 16x16 patches, results in a sequence length of 64x64 (4,096). For ViT-B, with 86M parameters (Dosovitskiy et al., 2021), this corresponds to a parameter-to-sequence length ratio of approximately 20,000.

While methods like LoRA effectively reduce the number of trainable parameters, they do not address activation gradients directly. Meaningful memory savings only occur when the initial layers of the model are frozen, allowing the optimizer to discard the corresponding activation gradients.

To test this hypothesis, we conducted an experiment using SAM trained on LIVECell (Edlund et al., 2021) data with a batch size of 1. We froze the entire image encoder except for a single block and observed the following results: unfreezing the first block of the image encoder reduced memory usage by approximately 500MB, while unfreezing the last block saved around 5GB compared to full fine-tuning.

These findings motivated the concept of Late PEFT. For the example in Late LoRA, LoRA adapters are applied only from a certain point onward in the image encoder, reducing memory usage by limiting activation gradient calculations, while still providing updates to the image encoder.

(Dettmers et al., 2023) suggest that once memory constraints are taken care of, performance can be further optimized by introducing additional trainable parameters through more low-rank adapters. Rather than limiting LoRA to the query and value matrices of the attention blocks, which is a common practice recommended by the original LoRA authors (Hu et al., 2022), we explore the application of LoRA to all linear layers within the attention blocks. We distinguish these approaches as "LoRA Classic" (LoRA-C) and "LoRA All" (LoRA-A) in Fig. 4.

We explore late PEFT methods by applying PEFT only from a specific attention block onward in the encoder, avoiding backpropagating gradients through the entire encoder, thereby reducing memory usage. We evaluate this approach using LoRA, QLoRA and full fine-tuning, applying each method to 50%, 25%, and 8% of the attention blocks. Our results in Fig. 13 and 14 show that the latter approach, applied to 50% of the blocks achieves the best trade-off between performance and memory efficiency.

## Appendix B. Implementation of QLoRA

For the implementation of QLoRA, we make use of bitsandbytes (`https://huggingface.co/docs/bitsandbytes/main`) for quantizing the linear layers. The implementation expects users to indicate linear layers to map to a lower precision (we reduce the precision down to 4 bit and freeze them, as training for this precision is a technical limitation in PyTorch). This necessitates converting the pretrained weights to the target precision (4 bit), initializing the quantized layers accordingly, and finally injecting LoRA to update the attention matrices. During training, the quantized layers are frozen, i.e. not updated through gradients. Their primary task is to improve efficiency; the LoRA matrices are learned at full precision. Another important finding is to avoid compound operations with tensors which require gradients, otherwise the PyTorch optimizer cannot trace gradients due to quantization in the computation graph, a known limitation of in-place operations in PyTorch. This contributed to the challenge of setting up a proper training implementation with quantization and mixed precision, combined with PyTorch optimizer specifics, in such a sensitive setup.

In inference, the approach to use the finetunined QLoRA model with quantized 4-bit layers, except for LoRA parameters of the attention matrices, did not work. After some empirical investigation, our analysis led us to introduce a new setup, where inferring with QLoRA model expects all layers to be initialized with the pretrained weights in the original precision, and replace the attention matrices with the learned LoRA weights. This resulted in meaningful improvements and comparable results to other PEFT methods in settings with small domain shifts.

## Appendix C. Implementation of Resource-efficient Finetuning

For microscopy, the choice of the 2 images for training was done manually by selecting images with a good representation of the overall task. CellSeg1 (Zhou et al., 2024) has shown that

the choice of image is crucial in this setting. Once these two images are selected, we use 1 image for training and the other exclusively for validation, and test our trained model on the entire test set, for consistency with other experiments. The same goes for medical imaging, however for 3D images, e.g. MOTUM and Mice TumSeg, we choose one plane with the foreground object annotations for training, and another from a different volume for validation. This step in particular is critical to avoid any negative bias, e.g. selecting an image with underrepresented / sparsely distributed cells or a 3D plane with an odd morphology of an organ.

We use CellSeg1 for benchmarking purpose with our data efficient experiments. Unlike CellSeg1, we improve SAM both for interactive and automatic segmentation. CellSeg1 only improves automatic segmentation, and is thus also not applicable for our medical experiments. Furthermore, we offer easy-to-use example scripts and a command line interface for finetuning. Our models are compatible with $\mu$SAM and the finetuning strategies without PEFT are directly compatible in several other annotation softwares, which is a clear limitation of CellSeg1 that can only be used within their tool.

## Appendix D. Evaluation Metric

We use the mean segmentation accuracy, following the definition of (Caicedo et al., 2019), to evaluate instance segmentation results for the microscopy datasets. It is computed based on true positives ($TP$), false negatives ($FN$), and false positives ($FP$), derived from the intersection over union (IoU) of predicted and true objects. Specifically, a $TP(t)$ is defined as the number of matches between predicted and true objects with an IoU above threshold $t$, $FP(t)$ correspond to the number of predicted objects minus $TP(t)$, and $FN(t)$ to the number of true objects minus $TP(t)$. The mean segmentation accuracy is computed over multiple thresholds:

$$\text{Mean Segmentation Accuracy} = \frac{1}{|\# \text{ thresholds}|} \sum_t \frac{TP(t)}{TP(t) + FP(t) + FN(t)}.$$

Here, we use thresholds $t \in [0.5, 0.55, 0.6, 0.65, 0.7, 0.75, 0.8, 0.85, 0.9, 0.95]$. For each dataset, we report the average mean segmentation accuracy over images in the test set.

We use the dice coefficient to evaluate segmentation results for medical imaging because the segmentation tasks we evaluate typically only contain a single object, or objects belonging to different classes, per image. It is defined as

$$\text{Dice Coefficient} = \frac{2 \cdot \sum p_i \, t_i}{\sum p_i + \sum t_i},$$

for per-pixel prediction values $p_i$ and per pixel target values $t_i$. For each dataset, we report the average dice coefficient over images in the test set.

## Appendix E. PEFT Evaluation

### E.1. Segmentation Quality

Fig. 3 presents a comparison of various PEFT methods across microscopy and medical datasets respectively. The PEFT methods are arranged from left to right in the order of

the number of trainable parameters, with full finetuning having the most parameters. Table 5 lists the number of parameters for each method. We evaluate the methods using training from the default SAM (Kirillov et al., 2023) and from the generalist $\mu$SAM models (Archit et al., 2025a), where we use either the electron microscopy or light microscopy model, and from MedicoSAM (Archit et al., 2025b) for medical imaging. The models are evaluated on automatic instance segmentation (AIS) for microscopy experiments, and point and box prompts for both microscopy and medical imaging experiments. We also report a simulated interactive segmentation setup, which we call the iterative prompting task, i.e. starting with either a single point ($I_P$) or a box prompt ($I_B$) for each object, and iterative rectification with additional points to improve the segmentation quality

As an overall trend, we observe that as the number of trainable parameters increases, the segmentation quality also tends to increase across the different datasets, demonstrating a trade-off between model complexity, capacity, and performance. The relative performance of the PEFT methods varies between datasets, indicating that the suitability of different PEFT methods may depend on the dataset's characteristics, such as complexity, size, and the model's initial performance. Despite this variation, Full Finetuning, Attention Tuning, and LoRA consistently show good performance. Two of the late PEFT methods, Late LoRA and Late FT, show some reduction in performance, but with relative minor differences to their "non-late" counterparts. (Late) QLoRA shows good performance for the domain specific foundation models ($\mu$SAM, MedicoSAM), but performs bad for default SAM, which has a larger domain gap.

To simplify these observations, we averaged the performance of the different PEFT methods over all datasets in Tabs. 1 (microscopy, SAM), 2 (microscopy, $\mu$SAM), 3 (medical imaging, SAM), and 4 (medical imaging, MedicoSAM). These averaged results validate our observations from above, highlighting that Attention Tuning, (Late) LoRA and Late FT show consistently good improvements. While some PEFT methods excel for specific tasks - such as Bias Tuning for MitoLab - they are not competitive with these other methods.

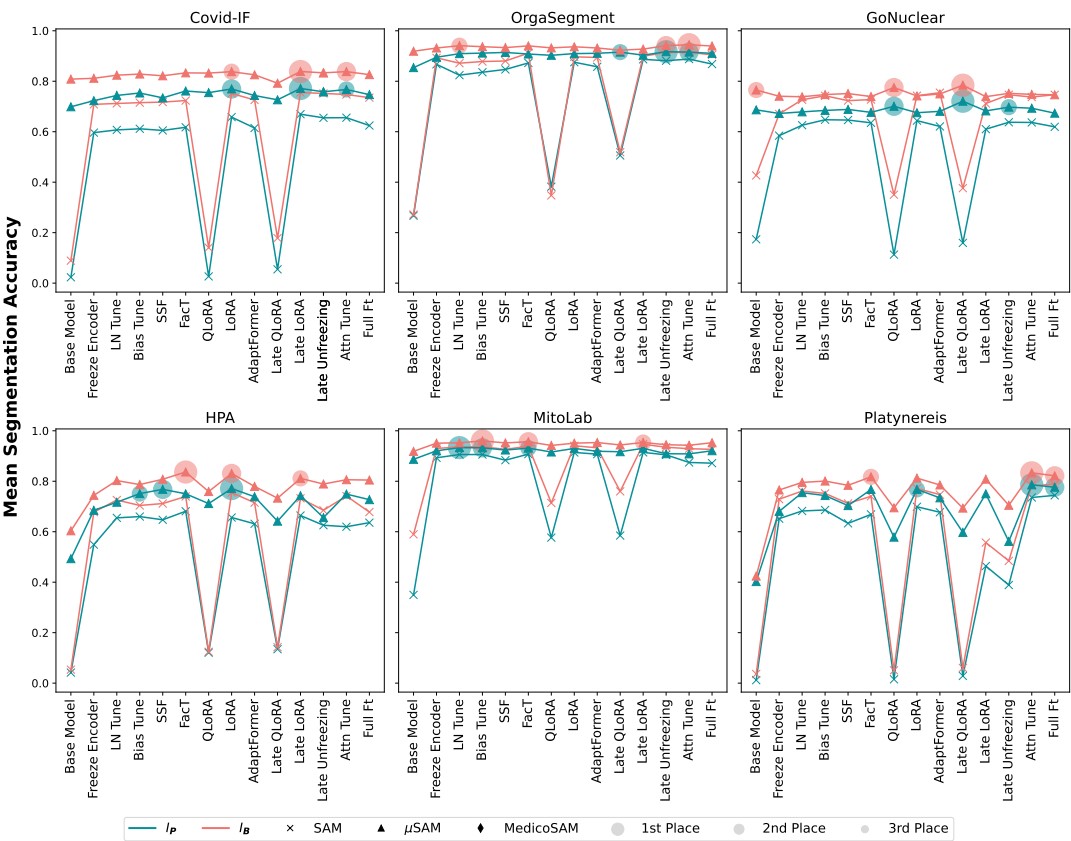

Figure 6: Results for iterative prompting on 6 different microscopy datasets, starting with point or box prompts. The accuracy is evaluated after 8 iterations of predicting and correcting prompts.

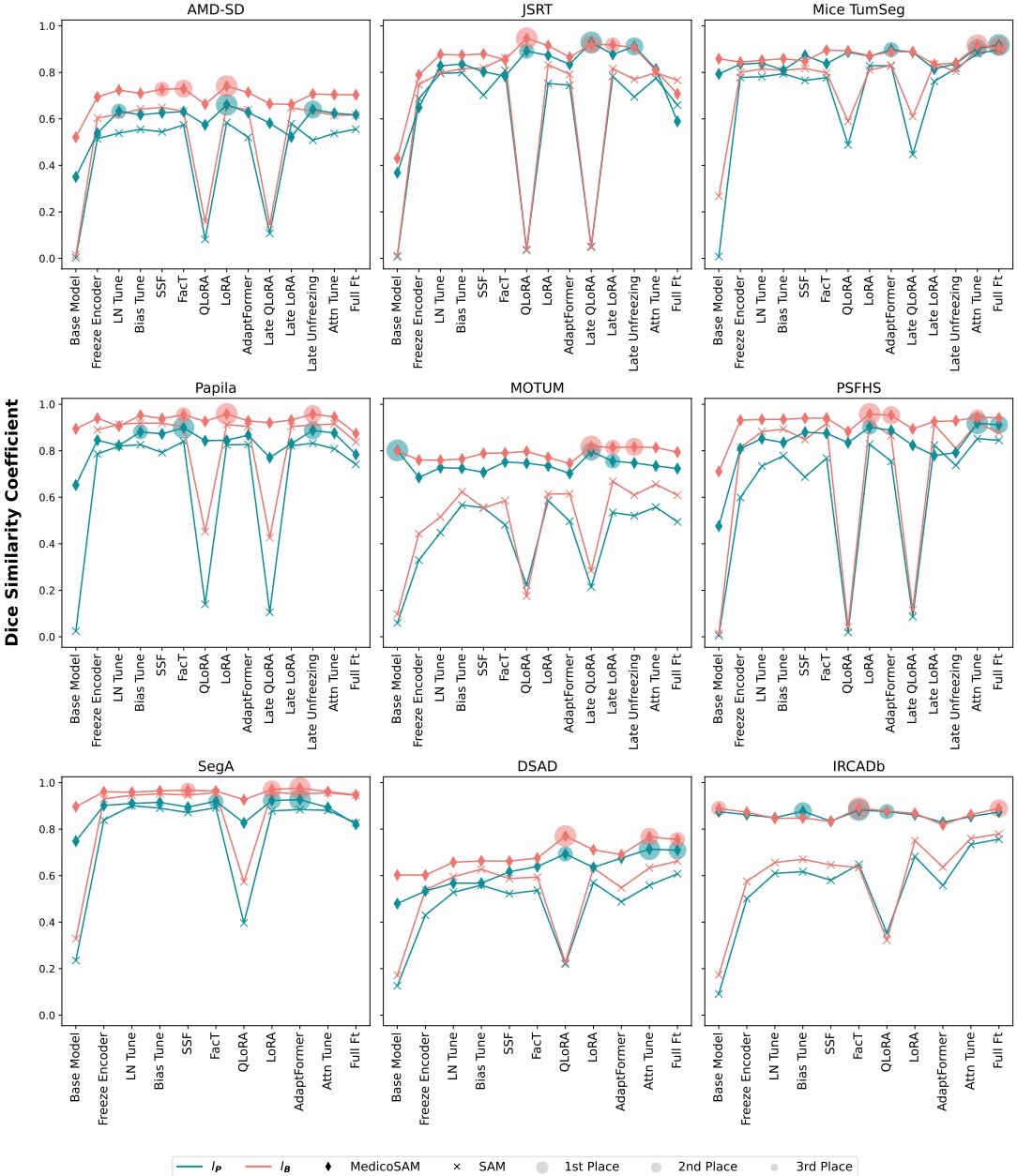

Figure 7: Results for iterative prompting on 9 different medical imaging datasets, starting with point or box prompts. The accuracy is evaluated after 8 iterations of predicting and correcting prompts. The late finetuning experiments are only run on the core 6 datasets.

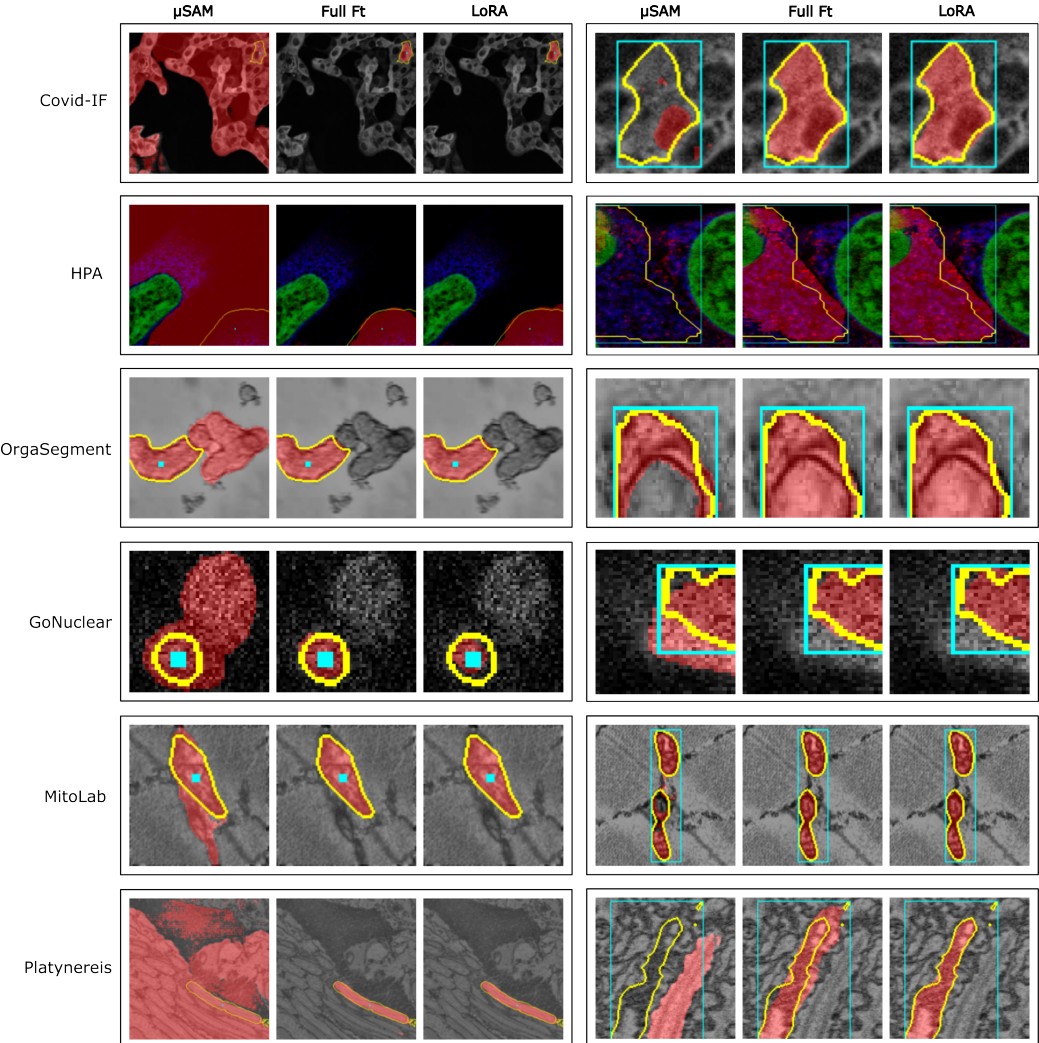

Figure 8: Qualitative comparison of interactive segmentation for the default $\mu$SAM model and the finetuned model with LoRA and Full Finetuning. For all three models ViT-B was used. The yellow outlines show the ground truth, red the model prediction and cyan shows the input prompts.

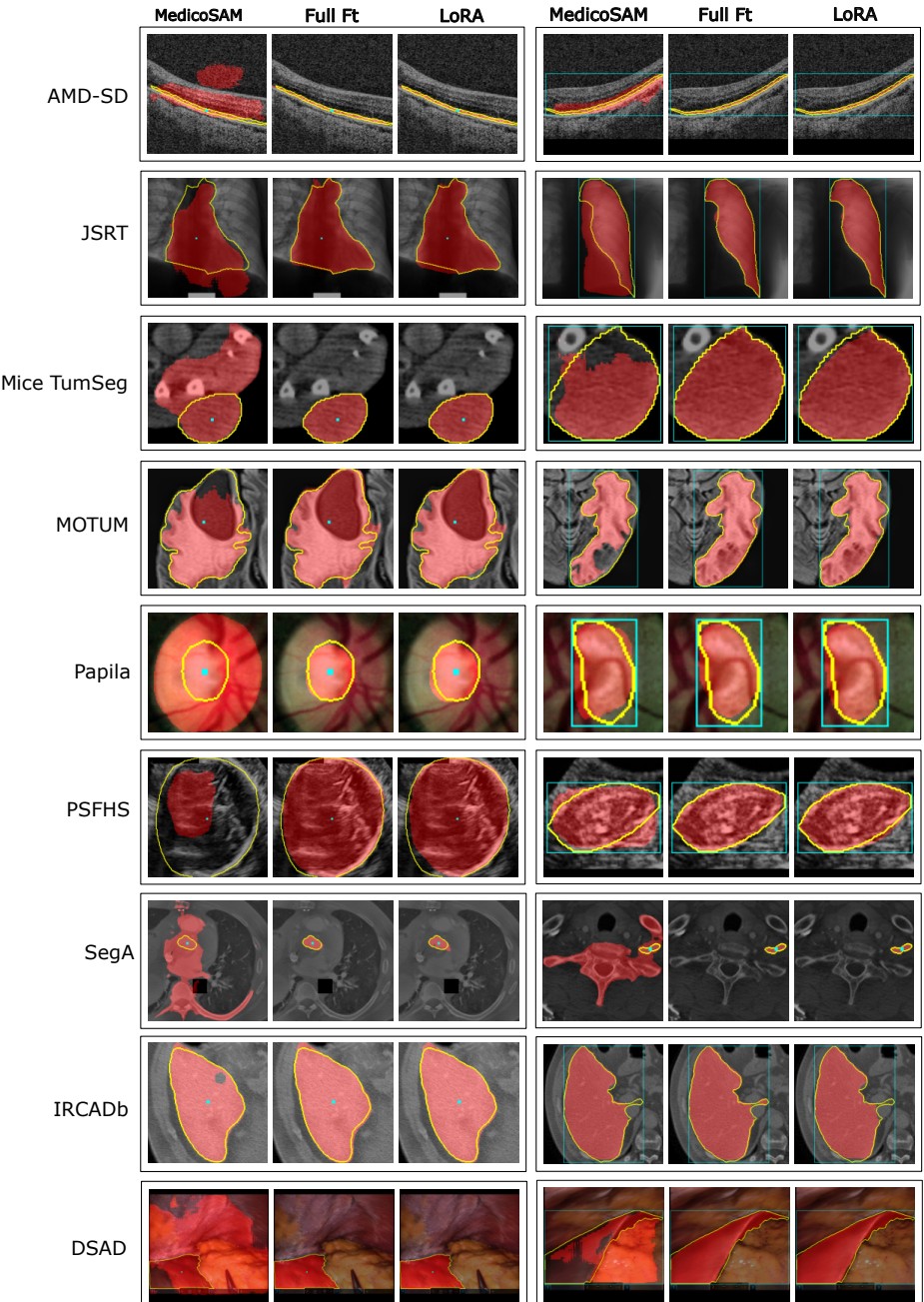

Figure 9: Qualitative comparison of interactive segmentation for the default MedicoSAM and the finetuned model with LoRA and Full Finetuning. For all three models ViT-B was used. The yellow outlines show the ground truth, red the model prediction and cyan shows the input prompts.

Table 1: The mean segmentation accuracy of different PEFT methods, trained from default SAM, for microscopy segmentation and averaged over all 6 datasets. (*): 1st Place; (**): 2nd Place; (***): 3rd Place.

| PEFT Method | AIS | Point | Box | $I_p$ | $I_b$ |
|---|---|---|---|---|---|
| Full FT | **0.352 | **0.303 | ***0.665 | 0.727 | 0.796 |
| Attn Tune | *0.352 | *0.308 | *0.673 | **0.735 | **0.809 |
| Late FT | 0.336 | ***0.300 | 0.615 | 0.675 | 0.753 |
| Late LoRA | 0.294 | 0.275 | 0.632 | 0.693 | 0.764 |
| Late QLoRA | 0.297 | 0.203 | 0.562 | 0.197 | 0.299 |
| AdaptFormer | 0.334 | 0.287 | 0.672 | 0.788 | 0.837 |
| AdaptFormer | 0.291 | 0.255 | 0.636 | 0.718 | 0.795 |
| LoRA | **0.343 | 0.296 | **0.667 | *0.741 | *0.810 |
| QLoRA | 0.281 | 0.218 | 0.529 | 0.206 | 0.288 |
| FacT | 0.328 | 0.270 | 0.644 | ***0.730 | ***0.797 |
| SSF | 0.291 | 0.245 | 0.625 | 0.710 | 0.779 |
| Bias Tune | 0.334 | 0.279 | 0.644 | 0.724 | 0.788 |
| LN Tune | 0.329 | 0.281 | 0.635 | 0.717 | 0.789 |
| Freeze Encoder | 0.309 | 0.267 | 0.620 | 0.690 | 0.768 |

Table 2: The mean segmentation accuracy of different PEFT methods, trained from $\mu$SAM, for microscopy segmentation and averaged over all 6 datasets. (*): 1st Place; (**): 2nd Place; (***): 3rd Place.

| PEFT Method | AIS | Point | Box | $I_p$ | $I_b$ |
|---|---|---|---|---|---|
| Full FT | **0.406 | *0.351 | ***0.696 | 0.793 | 0.849 |
| Attn Tune | *0.414 | **0.337 | **0.695 | **0.803 | **0.852 |
| Late FT | 0.371 | 0.315 | 0.639 | 0.752 | 0.829 |
| Late LoRA | ***0.393 | 0.333 | 0.675 | 0.790 | 0.844 |
| Late QLoRA | 0.334 | 0.295 | 0.621 | 0.746 | 0.807 |
| LoRA | 0.390 | ***0.336 | *0.702 | *0.804 | ***0.852 |
| QLoRA | 0.334 | 0.302 | 0.657 | 0.761 | 0.823 |
| FacT | 0.379 | 0.317 | 0.693 | ***0.799 | *0.854 |
| SSF | 0.324 | 0.274 | 0.670 | 0.789 | 0.842 |
| Bias Tune | 0.357 | 0.307 | 0.690 | 0.796 | 0.843 |
| LN Tune | 0.366 | 0.318 | 0.684 | 0.789 | 0.842 |
| Freeze Encoder | 0.330 | 0.268 | 0.655 | 0.763 | 0.824 |

Table 3: The dice coefficient of different PEFT methods, trained from default SAM, for medical segmentation and averaged over all 9 datasets. (*): 1st Place; (**): 2nd Place; (***): 3rd Place.

| PEFT Method | Point | Box | $I_p$ | $I_b$ |
|---|---|---|---|---|
| Full FT | **0.511 | ***0.685 | 0.709 | 0.779 |
| Attn Tune | *0.522 | *0.647 | *0.709 | **0.779 |
| Late FT | 0.405 | 0.687 | 0.685 | 0.756 |
| Late LoRA | 0.391 | **0.709 | ***0.717 | *0.797 |
| Late QLoRA | 0.288 | 0.560 | 0.169 | 0.272 |
| AdaptFormer | 0.335 | 0.648 | 0.678 | 0.753 |
| LoRA | ***0.424 | 0.694 | **0.726 | ***0.788 |
| QLoRA | 0.296 | 0.518 | 0.218 | 0.286 |
| FacT | 0.373 | 0.667 | 0.703 | 0.765 |
| SSF | 0.302 | 0.636 | 0.669 | 0.754 |
| Bias Tune | 0.367 | 0.647 | 0.710 | 0.772 |
| LN Tune | 0.335 | 0.653 | 0.684 | 0.749 |
| Freeze Encoder | 0.255 | 0.581 | 0.607 | 0.705 |

Table 4: The dice coefficient of different PEFT methods, trained from MedicoSAM, for medical segmentation and averaged over all 9 datasets. (*): 1st Place; (**): 2nd Place; (***): 3rd Place.

| PEFT Method | Point | Box | $I_p$ | $I_b$ |
|---|---|---|---|---|
| Full FT | **0.513 | 0.717 | 0.772 | 0.835 |
| Attn Tune | *0.516 | *0.733 | *0.815 | ***0.858 |
| Late FT | 0.406 | **0.731 | 0.803 | **0.859 |
| Late LoRA | 0.365 | 0.723 | 0.763 | 0.847 |
| Late QLoRA | 0.389 | 0.707 | 0.798 | 0.851 |
| AdaptFormer | 0.443 | 0.707 | ***0.805 | 0.842 |
| LoRA | ***0.485 | ***0.730 | **0.812 | *0.863 |
| QLoRA | 0.422 | 0.708 | 0.797 | 0.854 |
| FacT | 0.445 | 0.709 | 0.802 | 0.855 |
| SSF | 0.408 | 0.699 | 0.790 | 0.843 |
| Bias Tune | 0.414 | 0.692 | 0.785 | 0.841 |
| LN Tune | 0.409 | 0.697 | 0.781 | 0.835 |
| Freeze Encoder | 0.345 | 0.653 | 0.740 | 0.822 |

## E.2. Computational Efficiency

We measure the efficiency of PEFT methods, in terms of number of trainable parameters, training time per iteration and in total, and VRAM memory usage during training. Tab. 5 reports these measures for ViT-B and all PEFT methods, Tab. 6 and Tab. 7 report them for ViT-L and ViT-H respectively, for only a subset of PEFT methods. These results were obtained from finetuning default SAM on the LIVECell (Edlund et al., 2021) dataset, a large dataset with annotations for cell segmentation in phase-contrast microscopy. Note that we perform all experiments for segmentation quality with the smallest model, ViT-B, because it was shown e.g. in (Gu et al., 2025; Archit et al., 2025b) that using it does not have a noticeable impact compared to the larger models ViT-L and ViT-H for segmentation quality in biomedical imaging. However, we evaluate the computational efficiency for all model sizes, as this observation may not hold true for other domains, for which finetuning the larger models may be beneficial. We report efficiencies only for a single dataset, as these measurements are largely independent of the data specifics. The exception is the overall training time because we use early stopping. Specifically, we stop the training after 10 epochs without an improvement in the validation score. Hence, the overall training time depends on dataset characteristics. We report the training time across datasets in Fig. 10 and report the average over all datasets for each PEFT method in the same figure.

We see that all PEFT methods lead to a clear reduction in the number of trainable parameters, from 30% (Attn Tune) to less than ca. 5% (all others). However, this reduction in trainable parameters does not result in a clear reduction of memory requirements or training time per iteration. This fact is especially noticeable for ViT-B, for the larger models, especially ViT-H, the reduction in memory becomes a bit more pronounced. The exception is freezing the encoder, which clearly reduces the memory demand and also reduces training time per iteration, for all model sizes. With our new findings reported in Appendix A for Late PEFT, a future direction is quite clear for increasing efficiency of parameter efficient finetuning methods.

Table 5: Number of trainable parameters, training times, and allocated VRAM during training for different PEFT methods. Here, Train Time refers to the time until the best epoch is reached during training. Training times and memory are reported for finetuning default SAM with ViT-B on LIVECell. The number of parameters is shown as # Params w/o decoder | # Params with decoder. (*): selective PEFT; (**): additive PEFT

| PEFT Method | #Params [M] | Time / it [s] | Train Time [h] | Memory [GB] |
|---|---|---|---|---|
| Full FT | 93.7 \| 104.8 | 1.44 | 14.33 | 52.1 |
| Late FT | 46.6 \| 57.7 | 1.48 | 20.01 | 45.1 |
| Late LoRA | 6.42 \| 17.45 | 1.49 | 17.51 | 43.6 |
| Late QLoRA | 6.42 \| 17.45 | 1.53 | 21.31 | 45.8 |
| Attn Tune (*) | 32.5 \| 43.5 | 1.42 | 19.72 | 51.4 |
| AdaptFormer (**) | 5.3 \| 16.3 | 1.43 | 18.75 | 50.8 |
| LoRA (**) | 5.3 \| 16.3 | 1.44 | 17.59 | 51.2 |
| QLoRA (**) | 5.3 \| 16.3 | 1.50 | 18.34 | 51.1 |
| FacT (**) | 4.4 \| 15.4 | 1.45 | 19.60 | 51.2 |
| SSF (**) | 4.3 \| 15.3 | 1.49 | 16.12 | 53.3 |
| Bias Tune (*) | 4.2 \| 15.2 | 1.42 | 8.34 | 50.1 |
| LN Tune (*) | 4.1 \| 15.1 | 1.41 | 15.29 | 50.9 |
| Freeze Encoder (*) | 4.1 \| 15.1 | 1.24 | 16.79 | 35.5 |

Table 6: Number of trainable parameters, training times, and allocated VRAM during training for different PEFT methods. Here, Train Time refers to the time until the best epoch is reached during training. Training times and memory are reported for finetuning default SAM with ViT-L on LIVECell. The number of parameters is shown as # Params w/o decoder | # Params with decoder.

| PEFT Method | #Params [M] | Time / it [s] | Train Time [h] | Memory [GB] |
|---|---|---|---|---|
| Full FT | 312.3 \| 323.4 | 1.92 | 21.68 | 65.8 |
| Late FT | 79.68 \| 90.7 | 1.66 | 10.52 | 57.5 |
| LoRA | 7.2 \| 18.2 | 1.84 | 21.05 | 63.4 |
| QLoRA | 7.2 \| 18.2 | 2.06 | 20.18 | 63.3 |
| Freeze Encoder | 4.1 \| 15.1 | 1.37 | 18.55 | 36.3 |

Table 7: Number of trainable parameters, training times, and allocated VRAM during training for different PEFT methods. Here, Train Time refers to the time until the best epoch is reached during training. Training times and memory are reported for finetuning default SAM with ViT-H on LIVECell. The number of parameters is shown as # Params w/o decoder | # Params with decoder.

| PEFT Method | #Params [M] | Time / it [s] | Train Time [h] | Memory [GB] |
|---|---|---|---|---|
| Full FT | 641.1 \| 652.1 | 2.20 | 27.86 | 76.9 |
| Late FT | 122.2 \| 133.2 | 2.02 | 17.34 | 67.5 |
| LoRA | 9.3 \| 20.3 | 2.21 | 18.97 | 71.2 |
| QLoRA | 9.3 \| 20.3 | 2.49 | 19.43 | 69.7 |
| Freeze Encoder | 4.1 \| 15.1 | 1.51 | 14.29 | 37.8 |

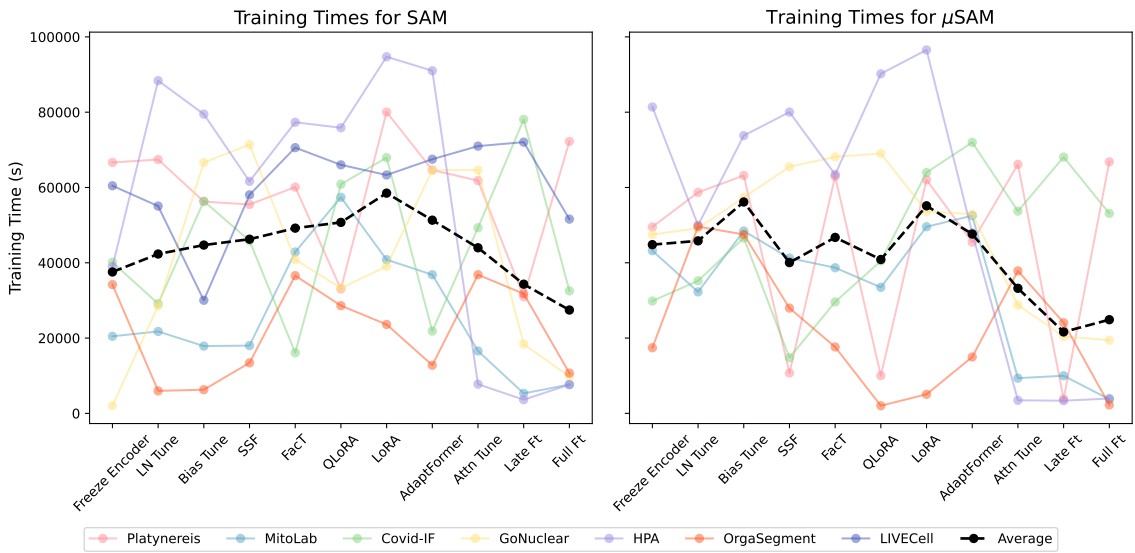

Figure 10: Time until convergence (early stopping) when training $\mu$SAM and SAM on the microscopy datasets, including LIVECell, for the PEFT methods. The dashed line in black represents the average train time taken by each PEFT method.

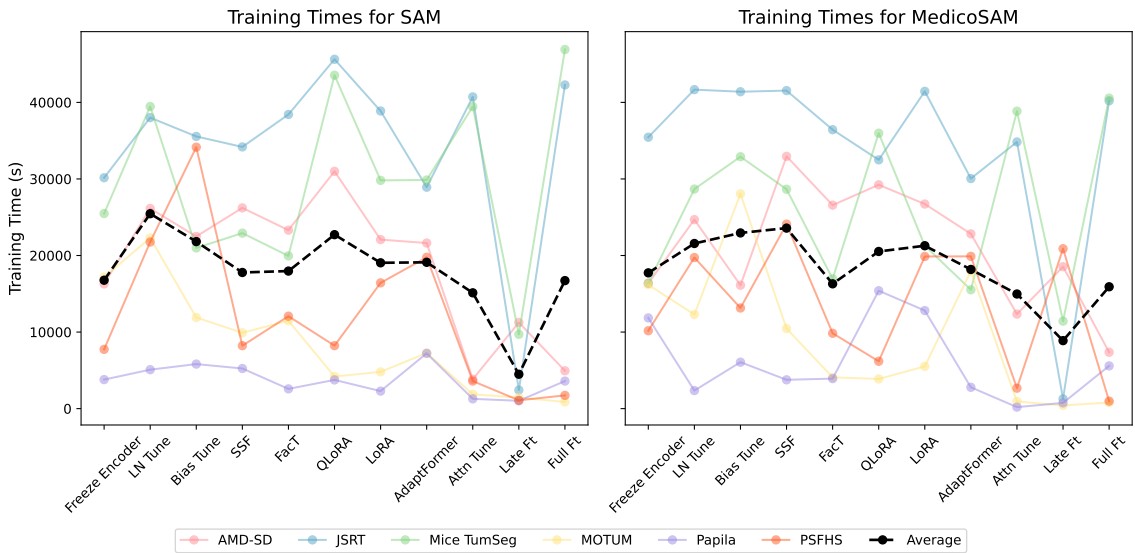

Figure 11: Time until convergence (early stopping) when training MedicoSAM and SAM on the 6 medical datasets for the PEFT methods. The dashed line in black represents the average train time taken by each PEFT method.

### E.3. PEFT Recommendations

After comparing the improvements in segmentation quality from different PEFT methods, compared to full finetuning (App. E.1) and their computational efficiency (App. E.2), we summarize our overall findings on PEFT for biomedical segmentation with SAM:

- PEFT methods that update the attention matrix weights, namely LoRA, Attention Tuning and their variants, lead to the most consistent improvement in segmentation quality, with improvements close to full finetuning.

- The newly introduced "Late" PEFT methods, which unfreeze parameters or insert adaptation layers only in the late ViT layers, yield almost as good results as their counterparts (i.e. adaptation applied to all ViT layers).

- PEFT **does not** lead to better segmentation quality than full finetuning, even when using only very small training datasets (see also App. F). Note that this observation is not trivial; for limited data one might expect advantages of less trainable parameters due to reduced overfitting.

- Only late PEFT yields meaningful improvements in computational efficiency during training, due to the fact that SAM finetuning is activation-bound rather than parameter bound (see also App. A).

Given these observations, we give the following recommendations for finetuning SAM for biomedical segmentation:

- With limited hardware resources (CPU): Use full freezing of the image encoder (Freeze Encoder); as this is the only viable option in this setting.

- With medium hardware resources (Consumer-grade GPU): Use Late Finetuning (Late FT), which provides the best quality-efficiency trade-off. Other Late PEFT approaches (e.g. Late LoRA) do not provide a meaningful improvement in efficiency and do not yield to better results.

- With high-end hardware resources (Server GPU): Use full finetuning, as PEFT does not provide any advantages in this setting.

While our recommendations where only tested for finetuning SAM for biomedical segmentation, we believe that they are also relevant for other tasks that involve ViT finetuning, as they will likely also be activation-bound rather than parameter-bound.

## Appendix F. Resource-efficient Finetuning

Fig. 4 shows the results for training on a single annotated image (with another used for validation). These results show that the performance of LoRA, when trained on a single image is highly dataset-dependent. For datasets like Platynereis (Vergara et al., 2021), which feature sparse instances within a single image, single-image training shows some limitations. Similarly, in the case of 3D datasets such as Platynereis, MitoLab, and GoNuclear, training on a single slice of one volume poses the additional challenge of selecting a suitable

slice. However, the performance achieved for MitoLab in this setting was notably strong. Here, when trained from the $\mu$SAM model, both LoRA and full fine-tuning achieved a mean segmentation accuracy of 0.54 when trained on a single image. Using all available images outperformed single-image training by only 6% for full finetuning and 2% for LoRA. As shown in Tab. 8, LoRA consistently reduces memory usage by approximately 500 MB compared to full fine-tuning. Thus, in resource constrained training settings, LoRA can be a practical solution, potentially enabling training scenarios that would otherwise be infeasible.

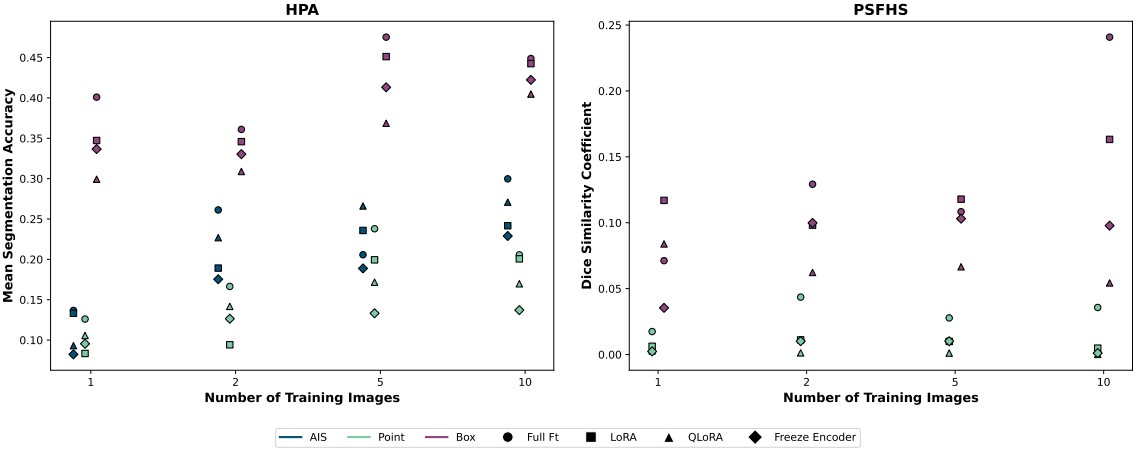

Figure 12: Data scaling experiments: SAM models are trained on $n$ training images and evaluated on the same test set.

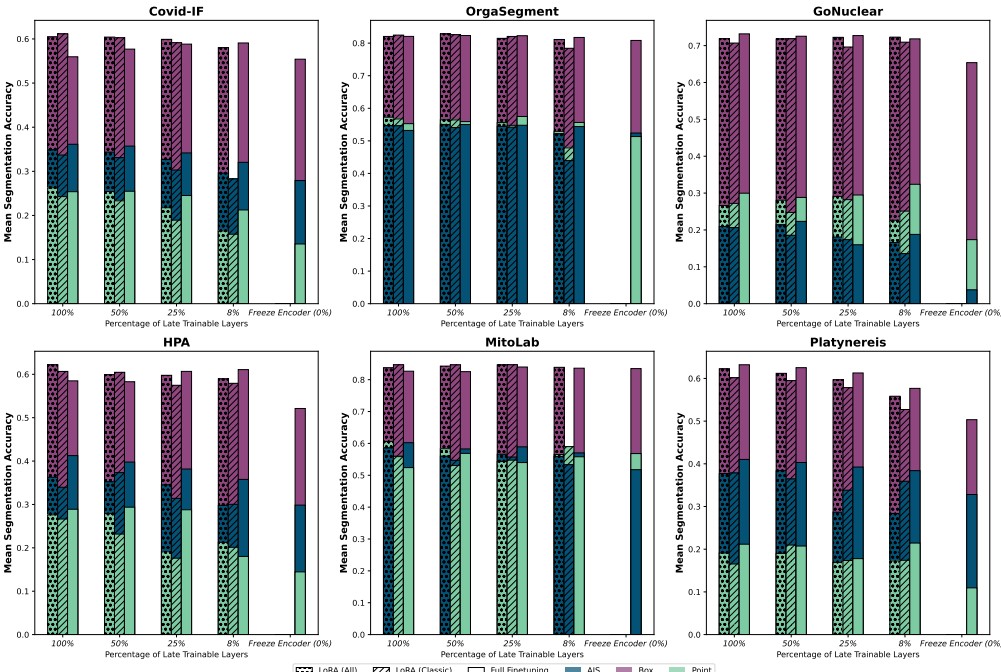

Figure 13: Quantitative results of $\mu$SAM models trained with late finetuning and late LoRA across different settings on 6 different microscopy datasets.

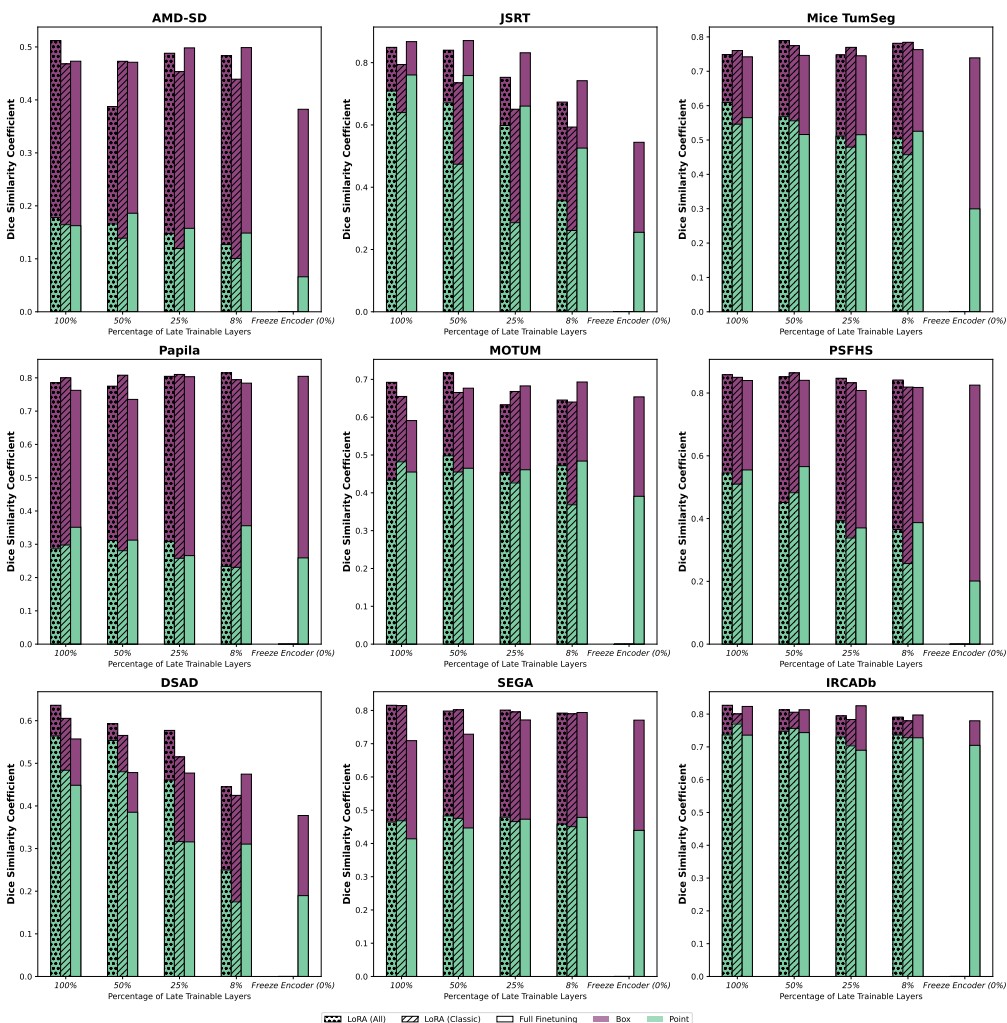

Figure 14: Quantitative results of MedicoSAM models trained with late finetuning and late LoRA across different settings on 9 different medical datasets.

Table 8: Allocated memory (in gigabytes) for each microscopy dataset during training on one image, comparing full fine-tuning, LoRA, QLoRA and freezing the encoder when training from SAM and $\mu$SAM.

| Allocated Memory [GB] | Covid-IF | Platynereis | MitoLab | OrgaSegment | GoNuclear | HPA |
|---|---|---|---|---|---|---|
| **SAM** | | | | | | |
| Full FT | 13.3 | 13.3 | 13.3 | 13.3 | 13.3 | 13.3 |
| Late FT | 9.7 | 9.6 | 9.6 | 9.7 | 9.7 | 9.7 |
| LoRA | 12.8 | 12.8 | 12.8 | 12.8 | 12.8 | 12.8 |
| QLoRA | 12.5 | 12.5 | 12.5 | 12.5 | 12.5 | 12.5 |
| Freeze Encoder | 5.8 | 5.8 | 5.2 | 5.8 | 5.8 | 5.8 |
| CellSeg1 | 7.0 | 7.7 | 8.4 | 11.3 | 9.6 | 8.0 |
| **$\mu$SAM** | | | | | | |
| Full FT | 13.3 | 13.3 | 13.3 | 13.3 | 13.3 | 13.3 |
| Late FT | 9.7 | 9.7 | 9.7 | 9.7 | 9.7 | 9.7 |
| LoRA | 12.8 | 12.8 | 12.8 | 12.8 | 12.8 | 12.8 |
| QLoRA | 12.6 | 12.5 | 12.6 | 12.5 | 12.6 | 12.5 |
| Freeze Encoder | 5.8 | 5.8 | 5.8 | 5.8 | 5.8 | 5.8 |
| CellSeg1 | 7.0 | 7.7 | 8.3 | 11.3 | 9.7 | 8.0 |

Table 9: Training time on one training image until convergence in minutes for each microscopy dataset, comparing full fine-tuning, LoRA, QLoRA and freezing the encoder when training from SAM and $\mu$SAM. For CellSeg1 the training time amounts to approximately 30 minutes, for 300 epochs, without early stopping.

| Train Time [min] | Covid-IF | Platynereis | MitoLab | OrgaSegment | GoNuclear | HPA |
|---|---|---|---|---|---|---|
| **SAM** | | | | | | |
| Full FT | 3.10 | 11.32 | 18.90 | 16.63 | 7.50 | 3.18 |
| Late FT | 4.31 | 15.97 | 11.58 | 8.37 | 15.39 | 2.20 |
| LoRA | 18.42 | 10.97 | 13.65 | 19.61 | 10.59 | 12.55 |
| QLoRA | 9.73 | 10.99 | 9.79 | 16.48 | 12.15 | 20.69 |
| Freeze Encoder | 2.38 | 19.81 | 12.77 | 16.56 | 10.73 | 9.38 |
| **$\mu$SAM** | | | | | | |
| Full FT | 5.41 | 8.33 | 2.35 | 1.18 | 5.57 | 1.04 |
| Late FT | 13.44 | 18.89 | 0.51 | 3.00 | 5.60 | 0.55 |
| LoRA | 5.34 | 18.48 | 3.39 | 5.55 | 2.26 | 3.11 |
| QLoRA | 13.20 | 25.36 | 2.32 | 7.04 | 3.54 | 2.14 |
| Freeze Encoder | 4.67 | 15.32 | 0.93 | 3.67 | 5.94 | 6.54 |

Table 10: Time per iteration in seconds on one training image for each microscopy dataset, comparing full fine-tuning, LoRA, QLoRA and freezing the encoder when training from SAM and $\mu$SAM.

| Train Time per Iteration [s] | Covid-IF | Platynereis | MitoLab | OrgaSegment | GoNuclear | HPA |
|---|---|---|---|---|---|---|
| **SAM** | | | | | | |
| Full FT | 0.74 | 0.71 | 0.69 | 0.71 | 0.69 | 1.27 |
| Late FT | 0.65 | 0.64 | 0.63 | 0.63 | 0.64 | 0.66 |
| LoRA | 0.71 | 0.73 | 0.68 | 0.69 | 0.67 | 1.25 |
| QLoRA | 0.73 | 0.73 | 0.73 | 0.73 | 0.69 | 1.31 |
| Freeze Encoder | 0.57 | 0.55 | 0.55 | 0.54 | 0.56 | 1.13 |
| **$\mu$SAM** | | | | | | |
| Full FT | 0.72 | 0.71 | 0.71 | 0.71 | 0.67 | 1.25 |
| Late FT | 0.65 | 0.63 | 0.62 | 0.60 | 0.61 | 0.66 |
| LoRA | 0.71 | 0.69 | 0.68 | 0.67 | 0.68 | 1.24 |
| QLoRA | 0.75 | 0.74 | 0.70 | 0.70 | 0.71 | 1.28 |
| Freeze Encoder | 0.56 | 0.54 | 0.56 | 0.55 | 0.55 | 1.12 |

# Appendix G. Ablation Study

## G.1. LoRA

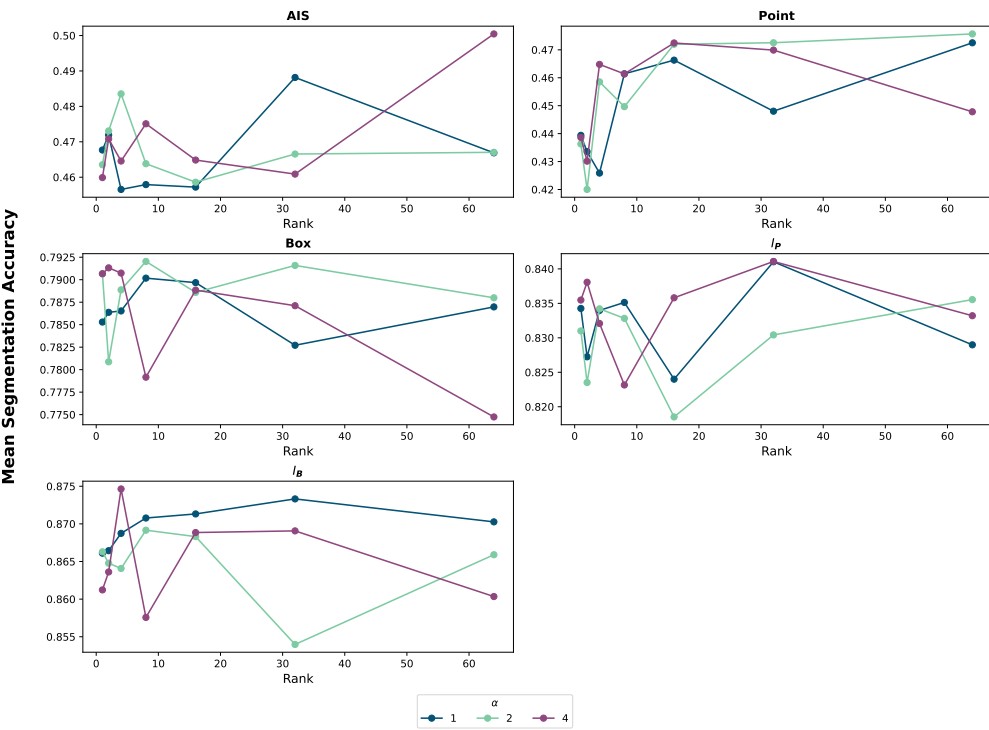

Figure 15: Inference results on OrgaSegment for LoRA with different combinations in $\alpha$ and rank. Results are shown in mean segmentation accuracy and evaluated across different tasks.

In this section, we conduct a series of experiments to investigate the influence of two key hyperparameters in LoRA: the rank and the scaling factor $\alpha$, as described in Section 2.1. Specifically, we aim to understand how these parameters impact segmentation accuracy and model performance.

(Hu et al., 2022) suggest that the scaling factor $\alpha$ behaves similarly to the learning rate, with both influencing the optimization process and model convergence. We explore this relationship through a series of experiments that vary $\alpha$ and rank in conjunction with the learning rate.

Figure 15 presents the impact of rank on segmentation accuracy for various values of $\alpha$. Intuitively, one might expect that increasing the rank would lead to improved accuracy, as it expands the parameter space. However, our experiments show that rank has only a marginal influence on iterative prompting. However, there is a noticeable improvement in the AIS and single point metrics as rank increases. Based on these findings, we choose a rank of 32 for all other experiments that use LoRA.

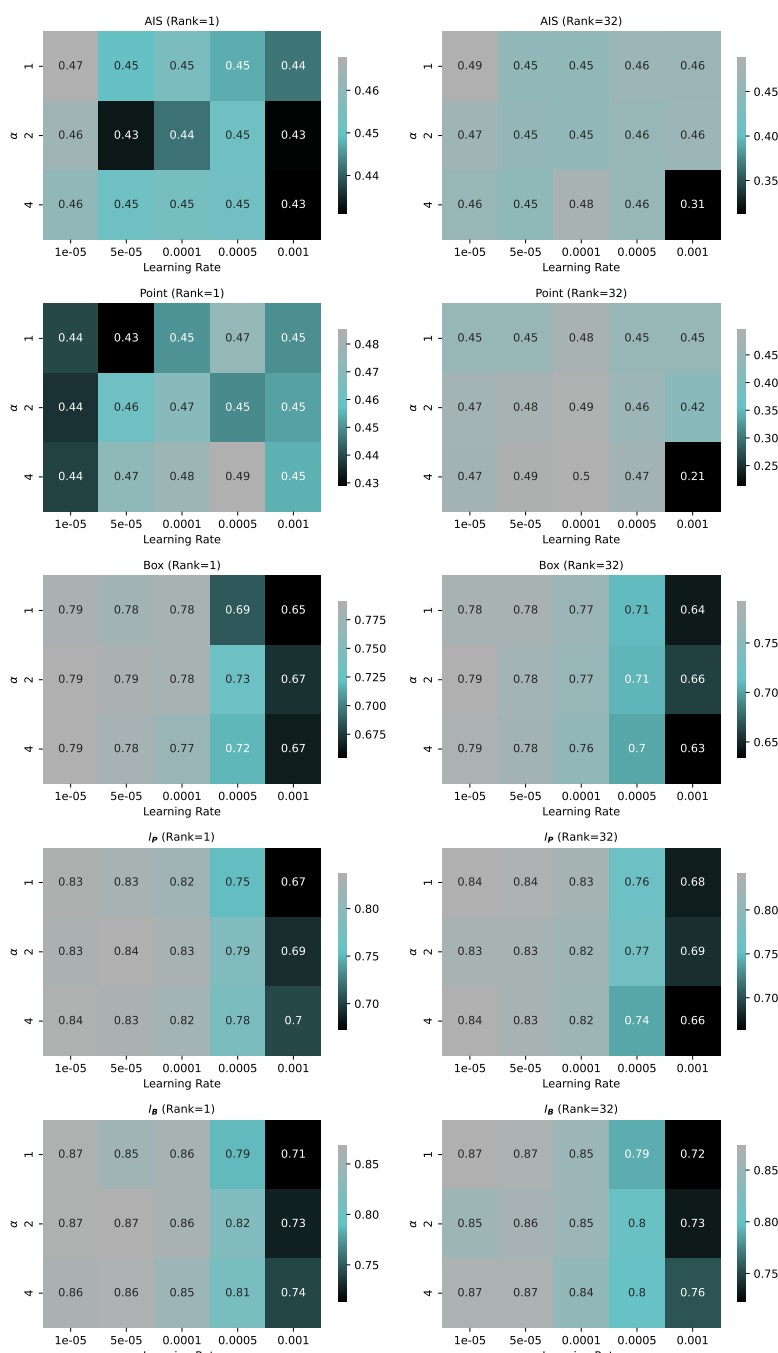

Figure 16: Inference results on OrgaSegment for LoRA with different learning rates and scaling factors $\alpha$. The colorbars represent mean segmentation accuracy.

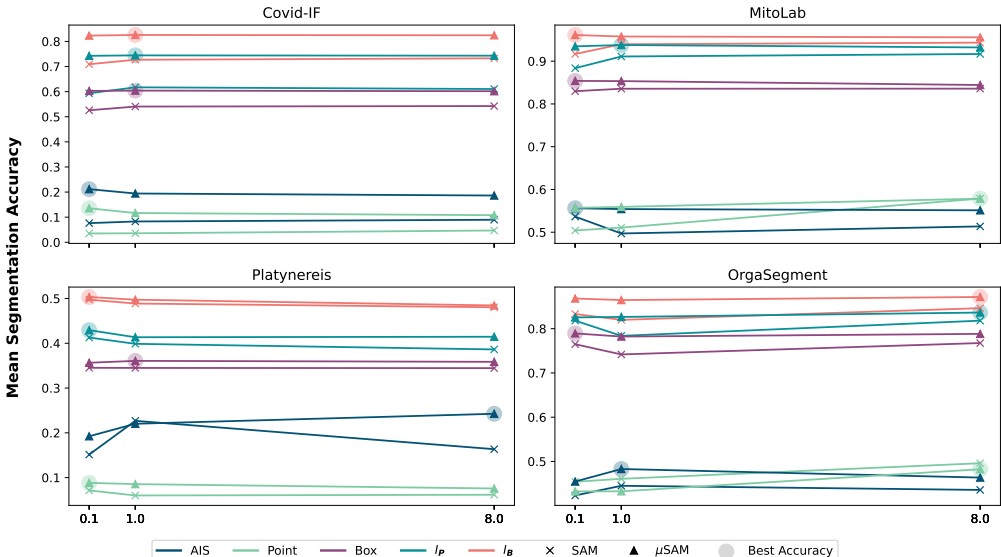

Figure 17: Inference results for LoRA with different scaling factors $\alpha$ on four datasets. Circles highlight he best results per dataset and task

Next, we investigate the relationship between the scaling factor $\alpha$ and the learning rate. Figure 16 illustrates that smaller learning rates tend to yield better performance across various values of $\alpha$. From this experiment, we select a learning rate of $1 \times 10^{-5}$ for all further experiments, as it provides the best trade-off between convergence and accuracy.

Our findings also suggest that using both a high learning rate and a large $\alpha$ simultaneously is detrimental to performance, potentially leading to instability in optimization. However, the precise influence of $\alpha$ remains ambiguous, as its impact is not as pronounced as that of the learning rate.

To further investigate the role of $\alpha$, we run experiments on four different datasets, as shown in Figure 17. The optimal value for $\alpha$ varies across datasets, and no consistent pattern emerges. However, we observe that using $\alpha = 1$ does not harm performance in any case, and we recommend using this default value for practical applications. In general, the scaling factor $\alpha$ appears to have a limited impact on the model's performance, and tuning it is not always necessary for achieving good results. We conclude:

- Rank has a marginal effect on iterative prompting but improves AIS and weak prompt performance, so we opt for rank 32.

- Learning rate is a more critical factor, with smaller learning rates being more favorable. A high learning rate and high $\alpha$ should be avoided.

- The optimal $\alpha$ value is dataset-dependent, but setting $\alpha = 1$ is a safe and effective choice across all datasets.

## G.2. AdaptFormer

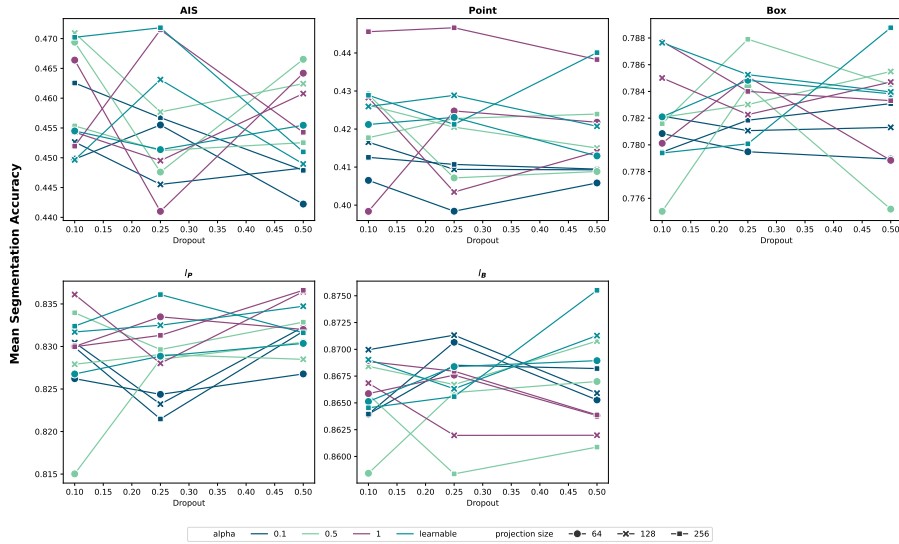

Figure 18: Inference results on OrgaSegment for different projection sizes and scaling factors and dropout values for AdaptFormer.

To find the best hyperparameters for AdaptFormer a grid-search was run on 3 parameters. The scaling factor, which scales the output of the Adapter module and therefore, similarly to LoRA, balances the impact of the features from the frozen branch with the task-specific features from the tunable parameters. Secondly, we try different projection sizes, meaning the middle dimension of the AdaptFormer module, which controls the number of learnable parameters that are introduced by this finetuning method. Lastly we introduce an optional dropout layer, between the down and up projection of the AdaptFormer branch.

Fig. 18 indicates that the dropout factor does not exhibit a consistent pattern in its impact on the inference results. Considering this lack of clarity and the additional stochasticity introduced by dropout during inference, we recommend setting the dropout to None by default, effectively excluding the dropout layer altogether. Assuming that the dropout value does not significantly influence the results, we average across the different dropout settings to achieve a more stable analysis of the alpha values and projection sizes. This is visualized in a heatmap for various inference tasks in figure 19. The results suggest a slight preference for larger projection sizes. However, the performance margin is minimal. Thus, we recommend a smaller projection size of 64 to reduce the number of parameters, balancing performance and complexity.

For the alpha parameter, the best results are observed when it is either learned by the model or set to one. Notably, the alpha values learned by the model are consistently close to one. To maintain flexibility for diverse datasets, we suggest using alpha as a learnable

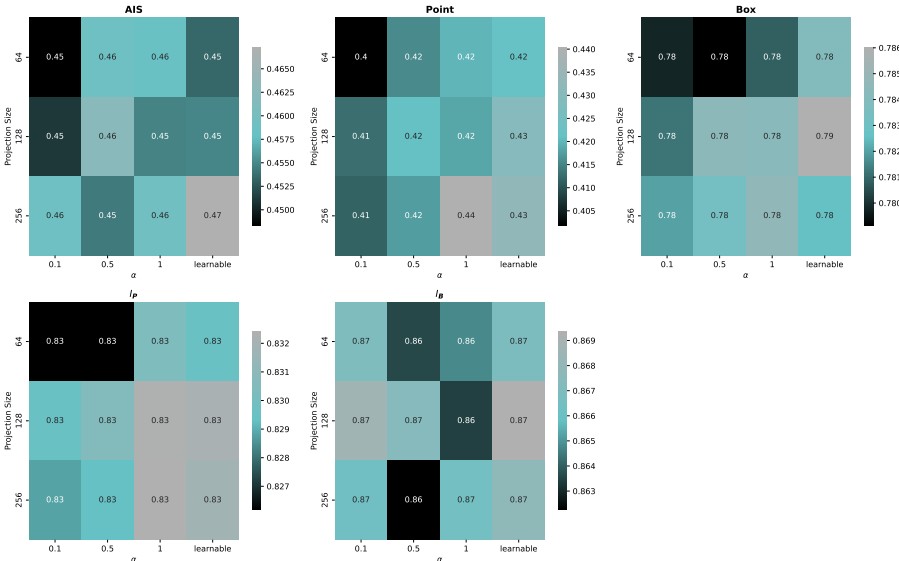

Figure 19: AdaptFormer inference results, trained on OrgaSegment for different scaling factors and projection sizes. The results are averaged over different dropout factor experiments.

parameter. However, if reducing the number of parameters is a priority, setting alpha to 1 is a suitable alternative.

## G.3. FacT

Table 11: Mean segmentation accuracy for dropout factor and rank parameter search for FacT on LIVECell.

| | ais | ip | ib | point | box |
|---|---|---|---|---|---|
| **Dropout Factor (Rank = 4)** | | | | | |
| 0.25 | 0.386388 | 0.783651 | 0.828160 | 0.408460 | 0.646480 |
| None | 0.383379 | 0.779622 | 0.822782 | 0.400679 | 0.64233 |
| 0.1 | **0.389440** | **0.787556** | **0.830561** | **0.413252** | **0.649592** |
| 0.5 | 0.385142 | 0.779190 | 0.821817 | 0.397363 | 0.640852 |
| **Rank (dropout=0.1)** | | | | | |
| 1 | 0.379699 | 0.774329 | 0.822471 | 0.382696 | 0.630324 |
| 2 | 0.383812 | 0.775398 | 0.822041 | 0.387330 | 0.638050 |
| 4 | 0.389440 | 0.787556 | 0.830561 | 0.413252 | 0.649592 |
| 8 | 0.393855 | 0.783390 | 0.826328 | 0.415375 | 0.648752 |
| 16 | **0.398600** | **0.792422** | **0.833112** | 0.426964 | 0.655786 |
| 32 | 0.397353 | 0.782903 | 0.831328 | **0.431270** | **0.657162** |

To analyze the impact of hyperparameter tuning in the FacT parameter-efficient fine-tuning method, we conducted an extensive search over two parameters: dropout factor and rank, using the LIVECell dataset. The dropout factor controls an optional dropout layer in the forward function of the FacT methods. We tested dropout factors of 0.1. 0.25, 0.5 and no dropout, finding that a dropout factor of 0.1 achieved the best overall performance. It led to consistent improvements across all tasks. For rank, we observed that increasing its value improved performance up to a rank of 16, which achieved the highest results across most tasks. Further increasing the rank to 32 provided only marginal improvements for weak prompts, while for AIS and iterative prompting, rank 16 remain superior. We therefore recommend using rank 16 for this method.

