# OpenReview forum: "Parameter Efficient Fine-Tuning of Segment Anything Model for Biomedical Imaging"
_MIDL.io/2025/Conference — MIDL 2025 Poster_

### Official Review · Reviewer_GBzG · 2025-02-21

**Confidence:** 5
**Preliminary Rating:** 4
**Recommendation:** Poster
**Final Rating:** 4

**Summary:**

In this paper, the authors investigate parameter-efficient fine-tuning (PEFT) methods for adapting large segmentation foundation models—specifically Segment Anything Model (SAM) and its specialized variants (μSAM for microscopy and MedicoSAM for medical images)—to biomedical image segmentation tasks. They compare nine PEFT approaches, including LoRA, QLoRA, AdaptFormer, Factor Tuning (FacT), SSF, and selective tuning strategies (e.g., freezing different parts of the model or tuning only biases) across twelve diverse microscopy and medical datasets. The experiments span both automatic instance segmentation (e.g., a specialized decoder predicting boundary and center distance maps) and interactive segmentation (initial prompts plus iterative corrections), revealing how each method balances performance and resource usage. They also introduce an implementation of QLoRA for vision transformers, making 4-bit quantization feasible during training, and demonstrate that while certain PEFT methods can deliver near state-of-the-art segmentation, encoder-freezing stands out as the most resource-friendly approach with relatively minimal performance drops. Importantly, the authors present a practical workflow for quickly fine-tuning SAM with limited labeled data (often only one or two images), lowering the computational barriers for biomedical segmentation tasks.

**Strengths:**

1.	the comprehensive exploration of multiple PEFT strategies, effectively mapping out their accuracy–efficiency trade-offs on real-world biomedical datasets (microscopy and radiological images).
2.	The authors not only evaluate LoRA but also delve into QLoRA, AdaptFormer, and other additive or selective tuning methods, thereby expanding prior work that focused predominantly on LoRA.
3.	Their experiments are methodologically rigorous, covering both instance segmentation (with a specialized decoder) and SAM-style interactive segmentation, which is rarely examined in detail when discussing parameter-efficient fine-tuning.
4.	the results underscore how resource-friendly strategies like encoder-freezing and QLoRA can still achieve excellent performance, thus promoting wider adoption of large foundation models in labs or clinics with limited GPU resources.

**Weaknesses:**

1.	It provides less emphasis on the underlying reasons why certain approaches (e.g., LoRA) might systematically outperform or underperform others
2.	Some findings, like QLoRA working better when the domain gap is small (e.g., μSAM) but faltering for large domain gaps, are empirically noted yet not deeply analyzed.

**Detailed Comments:**

1.	See weakness
2.	Further ablations on whether partial freezing or LoRA for the decoder portion might yield additional savings, especially for tasks with small training sets, could prove valuable.
3.	Provide more results or guidelines on how LoRA’s rank and scaling factor interact with learning rate in large domain-shift settings (e.g., electron microscopy or certain MRI subdomains).

**Justification Of The Final Rating:**

The authors have addressed most of my concerns, though some remain unresolved—particularly, why QLoRA shows significantly lower performance when the domain gap is larger. Still, I recommend a weak accept.

**Justification Of The Preliminary Rating:**

In this paper, the authors investigate parameter-efficient fine-tuning (PEFT) methods for adapting large segmentation foundation models—specifically Segment Anything Model (SAM) and its specialized variants (μSAM for microscopy and MedicoSAM for medical images)—to biomedical image segmentation tasks. Due to the weakness mentioned, I recommend weak accept.

**Questions To Address In The Rebuttal:**

Why Does QLoRA Show Markedly Lower Performance with Larger Domain Gaps?

**Special Issue:**

No

---

> ### Author Response · Authors · 2025-03-08
>
> Thank you for the thorough review and positive assessment of our work.
>
> > R4: It provides less emphasis on the underlying reasons why certain approaches (e.g., LoRA) might systematically outperform or underperform others.
>
> We agree that it would be valuable to understand the differences between approaches, especially w.r.t. their variability across datasets. However, addressing this is out-of-scope for our contribution, as it would require investigation of domain shifts of evaluation datasets compared to the training data of the foundation models.
>
> > R4: Some findings, like QLoRA working better when the domain gap is small (e.g., μSAM) but faltering for large domain gaps, are empirically noted yet not deeply analyzed.
>
> Indeed. A more rigorous study would require quantifying the domain shift between datasets. This is out of scope for our current work, but we want to study this in the future, possibly using recent methods for domain shift analysis (https://arxiv.org/abs/2503.00450).
>
> > R4: Further ablations on whether partial freezing or LoRA for the decoder portion might yield additional savings, especially for tasks with small training sets, could prove valuable.
>
> This question is now addressed by our experiments on late PEFT / late LoRA, which we will extend to the full set of datasets (i.e. Fig. 3, Fig. 4) for the camera ready version.
>
> > R4: Provide more results or guidelines on how LoRA’s rank and scaling factor interact with learning rate in large domain-shift settings (e.g., electron microscopy or certain MRI subdomains).
>
> This is indeed an interesting question, but due to the large number of experiments we had to run for the revision and the very limited time we could not fully address it. Note that the investigation on the rank and scaling factor in the appendix (Figure 13 - 15) already provide a detailed ablation of LoRA parameters.
>
> > R4: Why Does QLoRA Show Markedly Lower Performance with Larger Domain Gaps?
>
> See answer above.
>
> We have addressed the question regarding efficiency / late freezing in the review. Unfortunately, addressing questions about domain gaps is out of scope for our current contribution. We will add this to the potential future work in the camera ready version, including references to other work that could be used for quantifying domain gap (see the reference above). We unfortunately forgot to add this to the discussion of the revised version, and cannot update it anymore due to the deadline, but will make sure to include it in the camera ready version.

---

### Official Review · Reviewer_ENPm · 2025-02-21

**Confidence:** 3
**Preliminary Rating:** 2
**Recommendation:** Poster
**Final Rating:** 2

**Summary:**

This paper explores the potential of parameter-efficient fine-tuning to enhance interactive (SAM-like) semantic segmentation models for microscopy and medical images. These models are enhanced in their interactive and automatic applications. A benchmark using several datasets and PEFT methods is carried out, which puts special attention to LoRA and its quantized version.

**Strengths:**

- The benchmark topic is timely and interesting.
- The evaluation includes several PEFT methods and many datasets for transfer in both microscopy and medical imaging.
- The code implementation is open-access, which allows reproducibility and further progress in the field.

**Weaknesses:**

**[Clarity]** **Adaptation data**: The authors claim to use one single image for adaptation in Figure 5 (Section 3.2); however, it is unclear how much data is employed for adaptation in Section 3.1. **Efficiency analysis**: The studies on the computational efficiency of PEFT are included in the Appendix and scarcely introduced in the main manuscript. These experiments are an important motivation for PEFT. The way the manuscript is written makes these studies almost hidden. **Efficiency results**: Tables 5 and 6 report these, but only for microscopy datasets. Also, the results vary largely depending on the size of the ViT, but the authors do not provide further explanation. For instance, Full FT is faster than PEFT in Table 5 and way slower in Table 6. Also, Memory usage is largely stable between PEFT and Full FT - so what is the benefit of using PEFT in the explored context regarding efficiency? Then, Figures 8 and 9 report the training times per dataset, but the presentation is hard to follow - on average, FullFT seems the fastest - the authors could report the average across datasets to provide a more direct intuition.

**[Contributions]** The authors claim 4 contributions. However, the 3rd contribution is not motivated in detail. For example, from a technical perspective, the challenge of implementing QLoRA for ViTs or the modifications it requires should be further elaborated. Also, at some point, the authors claim to use “our finetuning approach”/”a new approach for resource efficiency”. However, it remains unclear what method the authors refer to since, to my knowledge, no new adaptation mechanism has been proposed.

**[Technical soundness] Contributions**: one of the contributions is “a detailed ablation of hyper-parameters for several PEFT methods, including LoRA.”. However, the experimental setting uses validation data that can be used for hyperparameter tuning. Thus, the contribution of providing a general hyper-parameter study is not clear since these could be fixed using the validation set. **How hyper-parameters are fixed**: The authors introduce detailed studies, for example, in alpha and rank in LoRA (k=32). Based on the Appendix results, the evaluation is carried out with averages across tasks. However, this hyper-parameter might not be optimal for all tasks - hence, why not use validation data to fix it?

**[Missing studies] Automatic segmentation for medical models**: Although AIS is incorporated into microscopy images, it is unclear why it is excluded from medical benchmarks. **Data efficiency**: The exploration of data efficiency involves using one annotated image to train and another to validate. The authors of Dutt et al. 2024 explore a larger umbrella of data regimes for image classification. Including an additional number of shots for adaptation or data-efficiency evolution figures would be important to achieve more conclusive conclusions.

**[It's Hard to draw conclusions] Best PEFT**: The authors claim that “LoRA is the best overall.” However, AttnTune seems robust (Table 2, 3), especially considering the lack of specific hyper-parameters. The authors could elaborate on this aspect and specify why LoRA is the best overall. **Do we need PEFT?** In the final recommendations, PEFT is only recommended in one scenario: using small models and not in low-data regimes. **Dataset-dependency**: As mentioned by the authors, there exists a significant degree of variability in each study depending on the dataset. However, it is unclear which is the source of such variability, e.g., why some datasets require fine-tuning and others PEFT, and how to detect it a priori.

**Detailed Comments:**

- The sentence “Notably, they need fewer annotated data compared to other models, requiring as few as a single labeled image (Zhou et al., 2024).” seems a bit strong since it is based on an Arxiv paper which has not been peer-reviewed yet (please, correct me if I am wrong, but I could not find a published version.).
- The importance given to QLoRA seems exacerbated. It does not perform as well as LoRA, and the differences in efficiency (Table 5, 6) are relatively small.
- The resolution of the Figures is low, making it difficult to follow the conclusions. For example, line plots in Figure 3 might not be appropriate since the x-axis has no continuity. Radar plots might be an option.

**Justification Of The Final Rating:**

First, I would like to thank the authors for their response and acknowledge their dedication to updating the manuscript.

1. I feel my concerns regarding [Technical soundness] are solved—the argument of avoiding extensive hyper-parameter tuning during deployment to ensure efficiency is valid. I now understand that validation data is only used for early-stopping purposes. Also, the authors have included more fine-grained data ratios in the few-shot adaptation experiments, and the quality of the figures has been greatly improved. Thanks.

2. There are some weaknesses which I still maintain after the author's responses:

- **[Contributions]**: The contribution of providing a QLoRA implementation for ViTs is not strong enough. I have a similar opinion to that of Reviewer SGi4 in this regard. Additionally, the experiments do not showcase a great advantage of using QLoRA across many scenarios, which burdens the potential impact of such implementation, as stated by the authors: "Indeed, this gain is rather limited in our current experiments".

- **[Missing studies]**:  The motivation of not including fine-tuning for semantic segmentation in the medical imaging modalities is not strong enough. Even though such fine-tuning would not allow for interactive segmentation, this is neither the intended objective of such fine-tuning, which is in contrast to perform predictions without user-dependency.

- **[“our finetuning approach”/”a new approach for resource efficiency]** I am not sure if considering such setting as a novel approach, since in Zhou et al., 2024, which is part of the motivation of the presented work, they already use a extremely low amount of adaptation data. Additionally, if such setting is a contribution/proposal, it would require additional weight in the main manuscript, and not in the Appendix (which we are not required to consider when reviewing the manuscript according to the MIDL guidelines).

- **[It's Hard to draw conclusions]** Regarding AttnTune, I still find it a strong baseline. As the authors mention, the number of parameters is not the only key metric, but also the memory consumption. According to Table 5, it is similar for both LoRA and AttnTune. Hence, AttnTune is not relatively "very heavy" - and should merit a more detailed consideration. Regarding dataset dependency, the authors leave it out of the scope of this work (which makes sense). However, the lack of detailed guidelines on addressing such variability quantitatively weakens some claims, e.g., the contribution of "a detailed ablation of hyperparameters for several PEFT methods" (since the optimal configurations are dataset-dependent) or the suggestion of relying primarily on LoRA.

- **[Overall clarity]** I agree with 3Ajs regarding the limited clarity and exposition of the contributions, the methodology, and most importantly the main takeaways.

3. Regarding additional contributions/efficiency analysis. The authors have included new contributions (“late PEFT”), and efficiency analysis (“activation bounded/memory consumption”). These new topics are now crucial to the experimental/conclusions section, and were not included in the initial manuscript. I consider it inappropriate to include new contributions or shift the focus in the manuscript during the discussion/rebuttal phase, the purpose of which is to address concerns/doubts. In addition, an essential part of their explanations is provided in the Appendix (which is not mandatory for reviewers to consider), so the main manuscript is not self-contained.

Based on these observations, I maintain my initial “Weak Reject” score. The study's objective has potential, but it also requires further development. If the paper is not accepted, I encourage the authors to filter out key ideas more carefully in future submissions.

**Justification Of The Preliminary Rating:**

I appreciate the introduction of a benchmark with recommendations on PEFT for interactive segmentation model adaptation. Also, I understand the challenges of developing such evaluations. However, the clarity and exposition of the experimental section are limited (I want to check the opinion of other reviewers as well). Also, there exists large variability within the results obtained for each dataset. The usefulness of PEFT is not fully motivated, given the results showcased. The efficiency studies in terms of data seem limited, and the computational efficiency results are mostly exposed in the Appendix, even though they are key motivations for using PEFT. Also, there are some technical choices for hyper-parameter tuning that are not entirely motivated. I would appreciate it if the authors could clarify my doubts and explain the inter-dataset variability, as well as whether PEFT is motivated by the results obtained - which is the main focus of the work, motivating the relevance of PEFT for SAM.

**Questions To Address In The Rebuttal:**

See justification for the recommendation.

**Special Issue:**

No

---

> ### Author Response · Authors · 2025-03-08
>
> Thank you for the thorough and constructive review of our work.
>
> > R3: [Clarity] Adaptation data: The authors claim to use one single image for adaptation in Figure 5 (Section 3.2); however, it is unclear how much data is employed for adaptation in Section 3.1.
>
> The experiments in Section 3.1 are conducted on the full datasets described in 2.4, which includes the size of the datasets. We use a train/test split with typically 75-80% of training data and the rest for testing.
>
> > R3:  Efficiency analysis: The studies on the computational efficiency of PEFT are included in the Appendix and scarcely introduced in the main manuscript. These experiments are an important motivation for PEFT. The way the manuscript is written makes these studies almost hidden.
> R3: Efficiency results: Tables 5 and 6 report these, but only for microscopy datasets. Also, the results vary largely depending on the size of the ViT, but the authors do not provide further explanation. For instance, Full FT is faster than PEFT in Table 5 and way slower in Table 6. Also, Memory usage is largely stable between PEFT and Full FT - so what is the benefit of using PEFT in the explored context regarding efficiency? Then, Figures 8 and 9 report the training times per dataset, but the presentation is hard to follow - on average, FullFT seems the fastest - the authors could report the average across datasets to provide a more direct intuition.
>
> We have updated the results on computational efficiency, please see the general comment for details and check out Figure 5 and the Appendix Section “Efficiency of PEFT”. We agree that the discussion of efficiency was not detailed enough and lacking from the main manuscript in the initial version, but believe that we have sufficiently addressed this. In particular, our new contributions uncover an important observation for finetuning vision transformers (that they are activation bound rather than memory bound), which to our knowledge has not yet been discussed in any detail. Based on this observation we introduce the new late PEFT strategies.
>
>
> > R3: [Contributions] The authors claim 4 contributions. However, the 3rd contribution is not motivated in detail. For example, from a technical perspective, the challenge of implementing QLoRA for ViTs or the modifications it requires should be further elaborated.
>
> Thank you for pointing out this omission. We study QLoRA because it a priori promises strong efficiency gains, which we however don’t observe due to the specifics of ViT training (see previous answer). We have adapted the description of this contribution and have added a section detailing its implementation to the appendix: “QLoRA Implementation”.
>
> > R3: Also, at some point, the authors claim to use “our finetuning approach”/”a new approach for resource efficiency”. However, it remains unclear what method the authors refer to since, to my knowledge, no new adaptation mechanism has been proposed.
>
> We mean the method used for the experiments in section 3.2, where we finetune a SAM model with only 2 annotated images (one for training, one for validation). We have clarified this in the text and have added a new section about this approach to the appendix: “Implementation of resource efficient finetuing”.
>
> > R3: [Technical soundness] Contributions: one of the contributions is “a detailed ablation of hyper-parameters for several PEFT methods, including LoRA.”. However, the experimental setting uses validation data that can be used for hyperparameter tuning. Thus, the contribution of providing a general hyper-parameter study is not clear since these could be fixed using the validation set. How hyper-parameters are fixed: The authors introduce detailed studies, for example, in alpha and rank in LoRA (k=32). Based on the Appendix results, the evaluation is carried out with averages across tasks. However, this hyper-parameter might not be optimal for all tasks - hence, why not use validation data to fix it?
>
> We don’t fully understand this comment. Trying to address it: “Why not use validation data to fix the best hyperparameters?” This would imply training a much larger number of models (number of hyperparam settings x number of datasets), which would exceed the computational budget available for our study. More importantly, it would also not be practical for users, as it would imply the need for a sufficiently larger separate validation set, and many training runs to find the optimal hyperparameter settings. Instead, our experiments on these settings show that they are fairly robust, especially w.r.t. the LoRA rank, the hyperparameter of the most used PEFT method.

---

> > ### Author Response · Authors · 2025-03-08
> >
> > > R3: [Missing studies] Automatic segmentation for medical models: Although AIS is incorporated into microscopy images, it is unclear why it is excluded from medical benchmarks.
> >
> > Here, we follow the implementations of microSAM (https://www.nature.com/articles/s41592-024-02580-4) for microscopy data and MedicoSAM (https://arxiv.org/abs/2501.11734) for medical data. The former implements AIS for microscopy data because automatic instance segmentation is a fairly standardized task for this domain, e.g. segmenting cells or nuclei in light microscopy. In contrast, automatic instance segmentation is not the relevant task in most medical image data settings. Rather, automatic semantic segmentation is required. However, this task is much more diverse, ranging from organ and lesion segmentation in MRI or CT imaging to segmentation of parts of the eye in fundus imaging. This makes it more difficult to train a comprehensive model for automatic semantic segmentation, and MedicoSAM does not attempt to do this. Instead, it studies specific finetuning for downstream semantic segmentation tasks, without preserving the capacity for interactive segmentation. Consequently, we only study fine tuning for interactive segmentation in medical imaging.
> >
> >
> > > R3: Data efficiency: The exploration of data efficiency involves using one annotated image to train and another to validate. The authors of Dutt et al. 2024 explore a larger umbrella of data regimes for image classification. Including an additional number of shots for adaptation or data-efficiency evolution figures would be important to achieve more conclusive conclusions.
> >
> > We have added a new experiment to study how the segmentation quality changes with additional images (1-10) for 2 datasets, one medical and one microscopy. You can find these results in Figure 12 (appendix).
> >
> > > R3: [It's Hard to draw conclusions] Best PEFT: The authors claim that “LoRA is the best overall.” However, AttnTune seems robust (Table 2, 3), especially considering the lack of specific hyper-parameters. The authors could elaborate on this aspect and specify why LoRA is the best overall.
> >
> > AttnTune is a “very heavy” approach to PEFT, as it updates all parameters of the attention layers, resulting in many trainable parameters. Compared to this, LoRA achieves a similar quality with a much smaller number of trainable parameters. Please also take into account our updated findings and discussion on efficiency of PEFT methods.
> >
> > > R3: Do we need PEFT? In the final recommendations, PEFT is only recommended in one scenario: using small models and not in low-data regimes.
> >
> > The recommendations are now updated to also take into account our findings on late PEFT. Please refer to the updated Discussion section for details.
> >
> > > R3: Dataset-dependency: As mentioned by the authors, there exists a significant degree of variability in each study depending on the dataset. However, it is unclear which is the source of such variability, e.g., why some datasets require fine-tuning and others PEFT, and how to detect it a priori.
> >
> > These differences are likely due to different domain shifts of these datasets compared to the foundation model’s training data, combined with the overall difficulty of the task. Addressing these questions in a rigorous manner is out of scope for our current contribution, as it would require a whole separate methodology toolkit to study domain shift, which is still an active area of research. We have however added this point to the discussion on limitations and future work, and point to recent publications on domain shift (https://arxiv.org/abs/2503.00450) that may prove useful in this context.
> >
> > > R3: The sentence “Notably, they need fewer annotated data compared to other models, requiring as few as a single labeled image (Zhou et al., 2024).” seems a bit strong since it is based on an Arxiv paper which has not been peer-reviewed yet (please, correct me if I am wrong, but I could not find a published version.).
> >
> > Indeed this paper is only available as an arXiv version. However, please note that we can reproduce their findings in our microscopy experiments in section 3.2. Furthermore, there is other work that shows successful adaptation of SAM based only on a small set of annotated images, for example https://www.nature.com/articles/s41592-024-02580-4 for microscopy and https://arxiv.org/abs/2404.09957  for medical imaging. We have chosen the Zhou et al. reference because it includes the most comprehensive experiments on training based on limited data.

---

> > ### Author Response · Authors · 2025-03-08
> >
> > > R3: The importance given to QLoRA seems exacerbated. It does not perform as well as LoRA, and the differences in efficiency (Table 5, 6) are relatively small.
> >
> > We study QLoRA because of its promise of better efficiency. Indeed, this gain is rather limited in our current experiments. However, note that this may change once we investigate using “Late QLoRA”, similar to our “Late LoRA” implementation, which is plausible due to the fact that VRAM usage is dominated by activations rather than trainable parameters in our settings (see the overview of the revision in the beginning). In any case, we now clarify that QLoRA efficiency gains are limited in the current settings.
> >
> > > R3: The resolution of the Figures is low, making it difficult to follow the conclusions. For example, line plots in Figure 3 might not be appropriate since the x-axis has no continuity. Radar plots might be an option.
> >
> > We have updated the plots to make them easier to read. In particular we reduced the number of quantities reported in the main paper and moved reporting of iterative prompting performance to the appendix. We tried using radar plots, but the results were not legible due to the different magnitudes of metrics for different evaluation modes.
> >
> > > R3: The usefulness of PEFT is not fully motivated, given the results showcased. The efficiency studies in terms of data seem limited, and the computational efficiency results are mostly exposed in the Appendix, even though they are key motivations for using PEFT.
> >
> > We have addressed this with the updated introduction, please refer to the general comment and the new version of the introduction for details.
> >
> > > R3: Also, there are some technical choices for hyper-parameter tuning that are not entirely motivated. I would appreciate it if the authors could clarify my doubts and explain the inter-dataset variability, as well as whether PEFT is motivated by the results obtained - which is the main focus of the work, motivating the relevance of PEFT for SAM.
> >
> > We now have addressed all these points in our revision and strongly believe that our work is suitable for MIDL.

---

> > > ### Comment · Reviewer_ENPm · 2025-03-14
> > > **Thanks for the responses.**
> > >
> > > I would like to thank the authors for their efforts to provide detailed answers to the reviewers' concerns. I have no doubts or additional clarifications needed.

---

### Official Review · Reviewer_SGi4 · 2025-02-27

**Confidence:** 4
**Preliminary Rating:** 2

**Summary:**

Vision foundation models, such as SAM, show a great potential to adapt to other downstream tasks. Thus, parameter-efficient finetuning (PEFT) is strongly relevant for this application. In this paper, authors contribute the first comprehensive study of PEFT for SAM applied to biomedical segmentation by evaluating 9 PEFT methods on diverse datasets.

**Strengths:**

1)  A clear writing and well-organized structure: Authors provide a good manuscript with clear enough details.
2)  The benchmark creation of PEFT methods on diverse datasets is a quite interesting topic.

**Weaknesses:**

1）Authors aim to unleash the power of SAM to resolve the medical image segmentation. For the title design, authors had better emphasize the medical domain, since there have been several papers with very similar titles, such as “Convolution Meets LoRA: Parameter Efficient Finetuning for Segment Anything Model”, “Parameter Efficient Fine-tuning via Cross Block Orchestration for Segment Anything Model”.
2）The implementation of QLoRA for ViTs cannot be viewed as an important contribution of a technical paper.
3）Authors simply mix up implementations from other groups, including implementations of QLoRA, procedures of interactive segmentation and instance segmentation. Besides, no novel designs or ideas are proposed. Sadly, I think those technical contributions cannot reach the standard of MIDL.
4）Also, to provide a thorough benchmark of PEFT methods, it will be more appropriate to select more challenging datasets, such as Totalsegmentors, BraTS, FLARE, EndoNerF, etc.

**Detailed Comments:**

This topic is quite interesting and highly requires further in-depth research. On the one hand, authors can conduct a methodology study by devising a PEFT method which outperforms other PEFT baselines. On that occasion, all these evaluation results in this paper can be quite useful as a given benchmark.
On the other hand, if authors are interested in the creation of thorough benchmarks on various datasets, I suggest a direct extension above this manuscript, by implementing evaluations on more challenging datasets, including Totalsegmentors, BraTS, FLARE, FETA, EndoNerF, AMOS, ATM, etc. After that, maybe a review journal paper can be prepared. A recent published can be referenced, “Trans-SAM: Transfer Segment Anything Model to medical image segmentation with Parameter-Efficient Fine-Tuning”.

**Justification Of The Preliminary Rating:**

This paper provides a quite crucial direction for medical image segmentation recently. However, there are main drawbacks on either the technical contributions or the limited evaluation on more extensive datasets. Based on a thorough evaluation, I think this manuscript is not solid enough to be published in its current version. Further work needs to be conducted.

**Questions To Address In The Rebuttal:**

Please refer to the section of detailed comments.

---

> ### Author Response · Authors · 2025-03-08
>
> Thank you for your correct summary of our work and the constructive review. We believe that our revision addresses all your concerns.
>
> > R2: 1）Authors aim to unleash the power of SAM to resolve the medical image segmentation. For the title design, authors had better emphasize the medical domain, since there have been several papers with very similar titles, such as “Convolution Meets LoRA: Parameter Efficient Finetuning for Segment Anything Model”, “Parameter Efficient Fine-tuning via Cross Block Orchestration for Segment Anything Model”.
>
> We agree that our original title was missing the reference to biomedical imaging. We have changed it to “Parameter Efficient Fine-Tuning of Segment Anything Model
> for Biomedical Imaging”  to address this point.
>
> > R2: 2）The implementation of QLoRA for ViTs cannot be viewed as an important contribution of a technical paper.
>
> We disagree with this statement. Adapting QLoRA to vision transformers was a difficult task, and should be counted as a technical contribution. We have added the section “Implementation of QLoRA” to the appendix to explain the implementation and its challenges.
>
> > R2: 3) Authors simply mix up implementations from other groups, including implementations of QLoRA, procedures of interactive segmentation and instance segmentation. Besides, no novel designs or ideas are proposed. Sadly, I think those technical contributions cannot reach the standard of MIDL.
>
> We believe that your comment shows a misunderstanding of what constitutes a technical contribution. We indeed build upon existing implementations of training methodology and PEFT methods. However, we had to implement all PEFT methods in a single framework, which is the only way to correctly evaluate them. Note that this work was motivated by the fact that many methods that adapt SAM to biomedical imaging, but none of them so far have actually evaluated the effectiveness and efficiency of it. Our work addresses this and thus marks an important contribution, only possible due to the implementation of existing methods in a shared framework. Further, please note that thorough evaluation of existing methods is considered a suitable contribution in all computer vision conferences. For example, a publication in last year’s MIDL conference evaluated PEFT methods for biomedical image analysis (https://openreview.net/forum?id=LVRhXa0q5r), though not for segmentation, and our work can be seen as an important complimentary contribution to their work. Finally, we now contribute a new approach to PEFT, namely “late LoRA” / “late freezing” based on the insights in the efficiency of PEFT, marking another important technical contribution of our work.
>
> > R2: 4) Also, to provide a thorough benchmark of PEFT methods, it will be more appropriate to select more challenging datasets, such as Totalsegmentors, BraTS, FLARE, EndoNerF, etc.
>
> We don’t use the datasets BraTS, FLARE, EndoNerf, AMOS and ATM (see below) in our benchmark because they were used for finetuning MedicoSAM (), which uses the SA-Med-2d dataset () that includes data from the aforementioned sources. Hence, using this data for adaptation would not be meaningful, as our initial model for medical segmentation was already trained on it. Totalsegmentator offers a very large and diverse datasets; integrating it into our experiments was not feasible in the short timeframe of the revision. However, please note that the datasets we use are diverse and challenging: we use data from optical coherence tomography (AMD-SD), inverted X-Ray (JSRT), micro-CT (Mice TumSeg), fundus imaging (Papila), Brain MIR (MOTUM) and ultrasound imaging (PSFHS). Only because these datasets may not be as known does not mean that they don’t represent challenging segmentation tasks; see Fig.  for quantitative examples. In addition, we have added experiments for three more datasets, SegA (aorta segmentation in CT), DSAD (organ segmentation in laparoscopy) and IRCADb (liver segmentation in CT), to the appendix, see Figure 7 and 9 for details. These 9 datasets now cover a wide range of challenging medical image segmentation tasks.

---

> > ### Author Response · Authors · 2025-03-08
> >
> > > R2: On that occasion, all these evaluation results in this paper can be quite useful as a given benchmark. On the other hand, if authors are interested in the creation of thorough benchmarks on various datasets, I suggest a direct extension above this manuscript, by implementing evaluations on more challenging datasets, including Totalsegmentors, BraTS, FLARE, FETA, EndoNerF, AMOS, ATM, etc. After that, maybe a review journal paper can be prepared. A recent published can be referenced, “Trans-SAM: Transfer Segment Anything Model to medical image segmentation with Parameter-Efficient Fine-Tuning”.
> >
> > We believe that all concerns raised by you have been addressed, including the comments on datasets, see the answers above. We strongly believe that our work is now suitable for MIDL. Thank you for pointing out the “Trans-SAM” publication, we have included it to the references for methods using PEFT for adapting SAM to medical imaging in the introduction. Please note that “Trans-SAM” does not study the effect of PEFT in any detail, further highlighting the importance of our work in providing an empirical basis for using PEFT for SAM.

---

### Official Review · Reviewer_3Ajs · 2025-03-01

**Confidence:** 3
**Preliminary Rating:** 2
**Final Rating:** 3

**Summary:**

The authors perform a comparison of 9 different parameter-efficient finetuning (PEFT) methods on the Segmen anything Model (SAM) and the µSAM  and MedicoSAM variants.
They evaluate these approaches on 12 datasets from Microscopy and Medical Imaging and aim to provide recommendations on how to use PEFT for efficient adaptation of SAM to medical segmentation tasks.

**Strengths:**

Strengths:
- The paper evaluates a large subset of the most used PEFT emthods.
- The paper aims to give practical recommendations on how to adapt segmentation models on a new task give computational contraints, which is  a realistic problem in the Medical Imaging field.
- The paper clearly explains the problem setting.

**Weaknesses:**

Weaknesses:
- Some models/approaches are selectively evaluated only on the Microscopy or only on the Medical Imaging datasets without a clear justification.
- The claim that PEFT does not lead to better results for limited training data is not backed up in the experimental evaluation.
- Even though the paper aims at providing recommendations for the use of PEFT, it is unclear what these recommendations are. The interpretation of the experimental results is also not obvious.
- It is unclear what the "workflow for efficient adaptation of SAM" is and how it differs from CellSeg1.

**Detailed Comments:**

Minor Suggestions:
- I would like to suggest making the colors in the line plots a bit easier to differentiate.

**Justification Of The Final Rating:**

Some of the raised concerns were well-addressed in the revision. The authors correctly pointed out that one of my concerns was based on a misunderstanding. However I believe that the clarity regarding the contributions, the methodology and the key takeaways could still be improved.

**Justification Of The Preliminary Rating:**

The paper tries to answer the important and interesting question of how to best perform PEFT for medical image segmentation and microscopy segmentation. Although the value of answering this question might be high, the paper currently fails to provide convincing answers.

**Questions To Address In The Rebuttal:**

The authors should clarify their interpretation of the results as well as their method evaluated in section 3.2. In general some claims do not seem to be backed up by the experimental evidence. Clarifying this or adding the necessary results to the paper would probably make sense.

---

> ### Author Response · Authors · 2025-03-08
>
> Thank you for your accurate summary of our contribution and the constructive review. We believe that our revision addresses all your concerns.
>
> > R1: Some models/approaches are selectively evaluated only on the Microscopy or only on the Medical Imaging datasets without a clear justification.
>
> We initially evaluated the resource efficient finetuning, using one image for training and one for validation, only on microscopy data. We have added the respective experiments for medical data (see updated Figures 3 and 4). We do not evaluate automatic semantic segmentation for medical images. This is due to the fact that semantic segmentation for medical imaging contains very diverse sets of tasks and pre-training a foundation model for this is challenging, see also our MedicoSAM work for details: https://arxiv.org/abs/2501.11734. In contrast, training of a foundation model for automatic instance segmentation in microscopy is feasible, because the task is much narrower (e.g. cell and nucleus segmentation in light microscopy). So we evaluate the automatic instance segmentation for this task. Apart from this exception, the experiments for microscopy data and medical imaging are now equivalent.
>
> > R1: The claim that PEFT does not lead to better results for limited training data is not backed up in the experimental evaluation.
>
> We strongly disagree with this statement. In almost all of our experiments “full finetuning”, i.e. updating all model parameters, performs on par or better than PEFT in terms of segmentation quality. This also includes the studies on resource efficient finetuning, which use limited training data of 2 images (one for training, one for validation), see Figure 4.
>
> > R1: Even though the paper aims at providing recommendations for the use of PEFT, it is unclear what these recommendations are. The interpretation of the experimental results is also not obvious.
>
> We have updated the discussion section to make the recommendation more clear. Please note that our new results on “late LoRA” / “late freezing” affect these recommendations. Briefly summarized, we now recommend to use full freezing for training on very limited resources (e.g. CPU), late LoRA for training with limited resources (e.g. a consumer-grade GPU) and full finetuning if a high-end GPU is available.
>
> > R1: It is unclear what the "workflow for efficient adaptation of SAM" is and how it differs from CellSeg1.
>
> We have added the section “Implementation of Resource-efficient Finetuning” to the appendix, which explains the workflow for efficient adaptation and differences to CellSeg1 in detail.
>
> > R1: The authors should clarify their interpretation of the results as well as their method evaluated in section 3.2. In general some claims do not seem to be backed up by the experimental evidence. Clarifying this or adding the necessary results to the paper would probably make sense.
>
> We have addressed all these points, see the answers to the previous questions. Given that we have addressed all your questions and have provided very valuable insights on the computational resource demand of parameter efficient finetuning in the revision, we believe that our work is very suitable for publication in MIDL, as it addresses an important topic in research on vision foundation models for medical imaging.

---

> > ### Comment · Reviewer_3Ajs · 2025-03-13
> >
> > Thank you for taking the time to address my comments.
> >
> > > We initially evaluated the resource efficient finetuning, using one image for training and one for validation, only on microscopy data. We have added the respective experiments for medical data (see updated Figures 3 and 4). We do not evaluate automatic semantic segmentation for medical images. This is due to the fact that semantic segmentation for medical imaging contains very diverse sets of tasks and pre-training a foundation model for this is challenging, see also our MedicoSAM work for details: https://arxiv.org/abs/2501.11734. In contrast, training of a foundation model for automatic instance segmentation in microscopy is feasible, because the task is much narrower (e.g. cell and nucleus segmentation in light microscopy). So we evaluate the automatic instance segmentation for this task. Apart from this exception, the experiments for microscopy data and medical imaging are now equivalent.
> >
> > Thank you for providing the additional experiments
> >
> > > We strongly disagree with this statement. In almost all of our experiments “full finetuning”, i.e. updating all model parameters, performs on par or better than PEFT in terms of segmentation quality. This also includes the studies on resource efficient finetuning, which use limited training data of 2 images (one for training, one for validation), see Figure 4.
> >
> > While full finetuning indeed performs better than PEFT in most of your experiments, there are no few-shot experiments or experiments isolating the number of datapoints used from other variables. It is therefore unclear how the presented experimental results substantiate the claim that PEFT does not lead to better results for limited training data.
> >
> > I would like to point out that the paper's main contribution would stand stronger if this claim were to be removed (or even better, substantiated with experimental evidence).
> >
> > > We have updated the discussion section to make the recommendation more clear. Please note that our new results on “late LoRA” / “late freezing” affect these recommendations. Briefly summarized, we now recommend to use full freezing for training on very limited resources (e.g. CPU), late LoRA for training with limited resources (e.g. a consumer-grade GPU) and full finetuning if a high-end GPU is available.
> >
> > Thank you for expanding on the discussion section with your recommendations.
> >
> > > We have added the section “Implementation of Resource-efficient Finetuning” to the appendix, which explains the workflow for efficient adaptation and differences to CellSeg1 in detail.
> >
> > I strongly suggest adding at least a rough sketch of the approach and its key novelty to the main paper.
> > The key novelty is still unclear from your explanation on the appendix. Further detail would help. If the main contribution is the easy to use interface, I would suggest clearly stating so.

---

> > > ### Author Response · Authors · 2025-03-14
> > >
> > > Thank you for engaging in the discussion.
> > >
> > > > While full finetuning indeed performs better than PEFT in most of your experiments, there are no few-shot experiments or experiments isolating the number of datapoints used from other variables. It is therefore unclear how the presented experimental results substantiate the claim that PEFT does not lead to better results for limited training data.
> > >
> > > We **do** provide these experiments. Figure 3 studies finetuning on all images of the respective training sets, for medical and microscopy datasets. Figure 4 does it for a single image. Figure 12 (appendix), studies finetuning on 1, 2, 5, or 10 images for 2 datasets, one microscopy, one medical. All these cases show that PEFT does not result in a better segmentation quality than full finetuning.
> > >
> > > > I would like to point out that the paper's main contribution would stand stronger if this claim were to be removed (or even better, substantiated with experimental evidence).
> > >
> > > We still strongly disagree with this comment, as we provide very strong evidence with experiments on over 12 datasets, for  training on full dataset (Figure 3), training on a single image (Figure 4), and training on 1-10 images (Figure 12, only 2 datasets).
> > > If you still disagree with our claim please specifically explain why the evidence we provide is insufficient.
> > > Please note that our claim is specific to our set-up (finetuning SAM for interactive and automatic segmentation) and cannot be generalized to other settings. We thought this was clear from our text, but are happy to clarify is in a camera-ready version.
> > >
> > > > I strongly suggest adding at least a rough sketch of the approach and its key novelty to the main paper. The key novelty is still unclear from your explanation on the appendix.
> > >
> > > Thank you for requesting the clarification here. We agree that adding some more information on the resource-efficient finetuning in the main text is desirable. Please keep in mind the page limit though, which makes it very hard to add substantial content to the main paper. We would try to add the following sentences to 3.2. in a camera ready version:
> > > "Briefly, our method for resource efficient fine-tuning implements training on two annotated images (one for training, one for validation). For it, the user has to annotate two representative images, and then apply our training method. Compared to a previous approach (CellSeg1), our method improves interactive and automatic segmentation with the model and yields models that may be used within other software based on SAM. See Appendix C for details."
> > > (or a similar formulation)
> > >
> > > > Further detail would help. If the main contribution is the easy to use interface, I would suggest clearly stating so.
> > >
> > > We would also improve this appendix section in a camera-ready version, to clearly state the contributions. In comparison to prior work, mainly CellSeg1, these contributions are:
> > > - Improvements of interactive and automatic segmentation. CellSeg1 improves the model only for automatic segmentation, which is also why we cannot compare to it for medical imaging.
> > > - Integration with popular and easy-to-use software (microSAM).
> > > - Possibility to export the so-trained models to a format compatible with other tools based on SAM under certain conditions. (This is only possible when using full finetuning, as LoRA adapters make models incompatible with other tools.)

---

> > > > ### Comment · Reviewer_3Ajs · 2025-03-14
> > > >
> > > > Thank you for providing further clarifications.
> > > >
> > > > It seems like the perceived lack of few-shot experiments was due my own misunderstanding of section 3.2. I concur that the experiments provide evidence for the claims about PEFT vs full fine-tuning. However I would suggest making the experimental setting used in section 3.2 more explicit.
> > > >
> > > > I believe that stated contributions should always be sufficiently explained in the main text of the paper. Details can be provided in the appendix, but the approach should be clear from the main text. Therefore I would still suggest to find a way to convey your approach in the main text as best as possible i the camera ready version.
> > > >
> > > > Due to your clarifications, I will adjust my rating upward.

---

> > > > > ### Author Response · Authors · 2025-03-14
> > > > >
> > > > > Thank you! We will do our best to extend the explanation of the experimental set-up for section 3.2 in the main paper under the given the page limit.

---

### Author Rebuttal · Authors · 2025-03-08

**Rebuttal:**

Dear Reviewers, dear area chair,

Thank you for taking our work under consideration & providing meaningful & extensive feedback. Based on these comments we have significantly updated our work in the following &:

- Study why we didn't see significant efficiency improvements due to most PEFT methods in our initial work. We find that this is due to the fact that our tasks are activation bounded rather than bounded by no. of trainable params. This distinguishes finetuning of SAM (& likely other ViTs) from LLMs. To best of our knowledge, this observation has not been done in the literature before. We have added the section “Efficiency of PEFT” to appendix to describe our findings in detail.
- Based on these findings, we introduce a new PEFT strategy, which we call “Late PEFT”, where trainable params are only inserted in late layers of the image encoder (a ViT). This enables more efficient finetuning, while providing clear quality improvements over the baseline of freezing full image encoder. We study this approach at the example of “Late LoRA” & “Late Freezing”, & have added a new figure, Fig5, to main text in order to study both computational efficiency & segmentation quality achieved with this approach. Note that we couldn't apply the approach to all datasets in Fig4 and Fig5 due to time constraints, but we will do so in the camera ready version. The results in Fig5 c) demonstrate the effectiveness of this approach and we don’t expect qualitative differences for other datasets.
- We now also perform resource-efficient finetuning (using 2 images for training, 1 for train set & 1 for validation) for medical images. Note that we only train for interactive segmentation, not for automatic segmentation in this case, thus we don't compare to CellSeg1 for medical.
- We have updated Fig3 & Fig4 to make them easier to understand for microscopy & medical results in a unified fashion: both figures now only contain results for segmentation with single point/box prompt &, automatic segmentation for microscopy. We have moved results for iterative prompting (IB, IP) to appendix (Fig6 for microscopy, Fig7 for medical, which also includes 3 addl. datasets). This makes main figures easier to read. Both figures now contain results for microscopy on the left & medical on right.
- We have added several other small experiments / explanations to appendix in order to satisfy individual reviewer requests.

All meaningful changes in the revised manuscript are highlighted in red.

**Supporting Material:**

/attachment/f50814cdeead0687c78c26f514ca9d89b4a10d03.pdf

---

### Author Response · Authors · 2025-03-13

Dear Reviewers,
we would be very grateful if you could respond to our comments to indicate if our revision satisfies your requests, or if there are any further questions / clarifications to address.

---

### Meta-Review · Area_Chair_UCMC · 2025-03-21

**Recommendation:** Accept (Poster)
**Confidence:** 4

**Metareview:**

- Overall evaluation

This paper presented a benchmark of param-efficient fine-tuning (PEFT) methods on microscopy image and general medical image analysis. The authors also introduced QLoRA, an extension of standard LoRA framework for this task. Some practical recommendations are also provided which can guide the utilization of PEFT methods.

- Strength

The benchmark of PEFT methods is very insightful to MIDL community. As PEFT has been widely investigated in general computer vision community, there is no systematic analysis on how PEFT can be adapted to medical imaging. This work is a good trial and the pratical recommendation is also promising.

- Weakness

As mentioned by the reviewers, the technical novelty is unclear in the manuscript. Also, the benchmark is also not thorough due to the page limits of MIDL. The authors should revise the manuscript to move the main focus to a benchmark/report rather than the technical contribution. My recommendation of this manuscript is still positive since it can provide an insightful view of PEFT methods for MIDL community.